

# Autoregressive neural Slater-Jastrow ansatz for variational Monte Carlo simulation

**Stephan Humeniuk[1,2⋆], Yuan Wan[1,3] and Lei Wang[1,3‡]**

**1** Institute of Physics, Chinese Academy of Sciences, Beijing 100190, China
**2** Department of Physics and Astronomy, Rutgers University, Piscataway, NJ O8854, USA
**3** Songshan Lake Materials Laboratory, Dongguan, Guangdong 523808, China

⋆ stephan.humeniuk@gmail.com , ‡ wanglei@iphy.ac.cn

## Abstract

Direct sampling from a Slater determinant is combined with an autoregressive deep neural network as a Jastrow factor into a fully autoregressive Slater-Jastrow ansatz for variational quantum Monte Carlo, which allows for uncorrelated sampling. The elimination of the autocorrelation time leads to a stochastic algorithm with provable cubic scaling (with a potentially large prefactor), i.e. the number of operations for producing an uncorrelated sample and for calculating the local energy scales like $\mathcal{O}(N_s^3)$ with the number of orbitals $N_s$. The implementation is benchmarked on the two-dimensional $t - V$ model of spinless fermions on the square lattice.

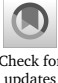

# 1 Introduction

Recently application of artifical neural networks to the variational simulation of quantum many-body problems [1] has shown great promise [2–12]. Variational Monte Carlo (VMC) simulations [13] with neural networks as an ansatz have in some cases surpassed established methods such as quantum Monte Carlo, which for fermions and frustrated spin systems in general has a sign problem, or tensor network states, which are limited by entanglement scaling. This success is due to the neural network's variational expressiveness [14], the ability to capture entanglement beyond the area law [14–16] and efficient sampling techniques.

    Most often for the VMC sampling a Markov chain with local Metropolis update is used [13]. This may result in long autocorrelation time and loss of ergodicity when the acceptance rate is too low, a limitation that is especially relevant for deep models [6,9,10] and in the simulation of molecular wavefunctions [17]. A technique for generating uncorrelated samples is the componentwise direct sampling, where the joint distribution $p_\theta(x)$ of a configuration $x$ of components is decomposed into a chain of conditional probabilties [18–23]

$$p_\theta(x) = \prod_{k=1}^{N} p_\theta(x_k | x_{<k}). \tag{1}$$

Here, $\theta$ denotes the variational parameters. The *components* $x_1, \ldots x_N$ of a configuration $x = (x_1, x_2, \ldots, x_N)$ are sampled iteratively by traversing the chain of conditional probabilities

$$p_\theta(x_1) \xrightarrow{x_1 \sim p_\theta(x_1)} p_\theta(x_2 | x_1) \xrightarrow{x_2 \sim p_\theta(x_2|x_1)} p_\theta(x_3 | x_1, x_2) \xrightarrow{x_3 \sim p_\theta(x_3|x_{<3})} \ldots, \tag{2}$$

and inserting the value of the sampled component into the next conditional probability. As a result, a sample drawn according to the joint distribution $x \sim p_\theta(x)$ and its (normalized)

probability $p_\theta(x)$ are yielded. Such autoregressive generative models, which are widely used in image and speech synthesis [24, 25], enjoyed several elegant applications in the physical sciences, namely to statistical physics [22], the reconstruction of quantum states with generative models [26], quantum gas microscopy [27], design of global Markov chain Monte Carlo updates [28] and variational simulation of quantum systems [11, 23, 29–31]. Direct sampling has also been employed in the optimization of tensor networks [32–34].

As long as the configuration components are spins that sit at fixed positions, a natural ordering in which the autoregressive property holds can be imposed easily. On the other hand, adapting the autoregressive approach to *indistinguishable* particles with a totally antisymmetric wavefunction, requires a number of modifications. The antisymmetry of the fermionic neural network wavefunction has been imposed in various ways: In Ref. [35] the antisymmetry was implemented directly as a symmetry [5] by keeping track of the sign changes due to permutation from a a representative configuration for a given orbit of the permutation group. Then no Slater determinant needs to be computed which results in a $\mathcal{O}(N^2)$ rather than $\mathcal{O}(N^3)$ scaling with the system size $N$ [35]. In Refs. [17, 36] the sign structure was encoded at the level of the Hamiltonian operator rather than the wavefunction by mapping fermionic degrees of freedom to spins via a Jordan-Wigner transformation. For completeness, it is worth mentioning that in the setting of first quantization, the deep neural networks FermiNet [37] and PauliNet [38, 39] have achieved remarkable success in *ab initio* simulations by applying a few generalized determinants [40] to multi-orbital wavefunctions of real space electron positions encoded as a permutation-equivariant neural network ansatz. Alternative first-quantized approaches aimed at replacing the costly $\mathcal{O}(N^3)$ determinant evaluation by a cheaper antisymmetrizer [41] scaling as $\mathcal{O}(N^2)$ appear to come at the price of reduced accuracy [42, 43].

By far the most commonly employed variational wavefunction in VMC for fermions [44, 45] is an antisymmetric Slater determinant (SD) [46] multiplied by a symmetric Jastrow correlation factor [47]

$$|\psi_\theta\rangle \sim \sum_x |x\rangle \mathcal{J}(x)\langle x|\psi_0\rangle, \tag{3}$$

where $\sim$ indicates that the wavefunction written in this form is in general not normalized. A famous example of a variational wavefunction of Slater-Jastrow form is the Laughlin wavefunction [48] describing quantum Hall states. The Jastrow factor $\mathcal{J}(x)$, which is diagonal in the local basis $\{x\}$, encodes complex dynamical correlations by altering the modulus of the amplitudes of basis states, however, it does not affect the nodal structure of the wavefunction, which is solely determined by the mean-field reference wavefunction $|\psi_0\rangle$. The latter is either a Slater determinant, or a Pfaffian or correlated geminal [49], which is an implicit resummation of a subset of Slater determinants. Neural VMC with an ansatz of the form of Eq. 3, where the Jastrow factor is represented by a neural network, has been explored for fermions [2, 4, 7] and also for spin systems [50], through a Gutzwiller projected mean-field wavefunction augmented by a Jastrow correlation factor. The Slater-Jastrow ansatz can be extended to incorporate static (i.e. multi-reference) correlations beyond a single Slater determinant [2, 4, 7, 51].

Here, we focus on lattice models, and we consider only the case where the reference wavefunction $|\psi_0\rangle$ is a single Slater determinant. Thus, static (i.e. multireference) correlations are not captured, which is an inherent limitation of the ansatz. The emphasis of this paper is on improving the sampling efficiency [52] by imposing the autoregressive property on both the Slater determinant [53–55] and the Jastrow factor so that uncorrelated sampling becomes possible. As illustrated schematically in Fig. 1, the conditional probabilities are interlaced into

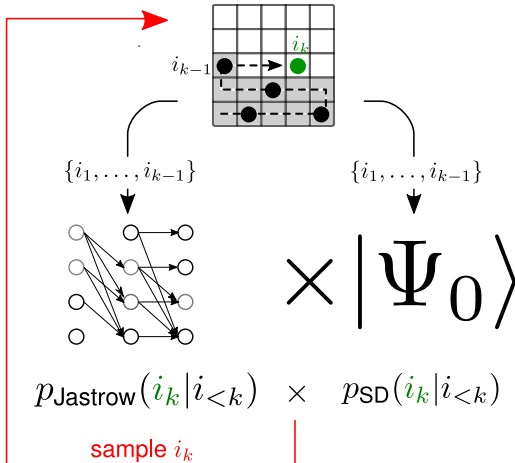

Figure 1: Combination of an autoregressive neural network for the Jastrow factor with an autoregressive Slater determinant (SD) into a Slater-Jastrow ansatz which allows direct sampling of many-particle configurations.

a combined autoregressive ansatz

$$\Psi_{SJ}(x) = \text{sign}(\langle x|\psi_0\rangle)\sqrt{\prod_{k=1}^{N_p} p_{\text{SD}}(i_k|i_{<k})}, \tag{4}$$

where $p_{SJ}(i_k|i_{<k}) = \mathcal{N} p_{\text{Jastrow}}(i_k|i_{<k}) p_{\text{SD}}(i_k|i_{<k})$, with normalization factor $\mathcal{N}$, and where the sign structure is determined solely by the Slater determinant $|\psi_0\rangle$. While generic *components* are denoted as $x_k$, we write $i_k \in \{1,\ldots,N\}$ and $n_{i_k} \in \{0,1\}$ if the components are particle positions or occupation numbers, respectively.

In simulations of lattice models with a Slater-Jastrow variational wavefunction, which includes orbital space VMC [56–59], the autocorrelation time increases as some power of the system size [56] so that eliminating autocorrelation would change the scaling of the method. Also bottlenecks due to low acceptance rates [17] are avoided by direct sampling. We note the related work of autoregressive neural-network wavefunctions for quantum chemistry applications [60,61] that have recently been proposed for VMC in the space of arbitrary excited Slater determinants ("full configuration interaction"). Our approach is different in that all correlations are handled by the Jastrow factor.

This paper is structured as follows: In Sec. 2 the building blocks of the autoregressive Slater-Jastrow ansatz are first introduced separately: a permutation invariant masked autoregressive deep neural network representing the (bosonic) Jastrow factor (Sec. 2.1.1) and the Slater sampler, which can sample unordered (Sec. 2.2.1) or ordered (Sec. 2.2.2) particle positions from the mean-field wavefunction. Sec. 2.3 discusses the issue of normalization which arises from the multiplication of the Jastrow factor and the Slater determinant and has important implications both for the probability density estimation of a sample and the calculation of the local energy of the combined ansatz. Sec. 2.5 is devoted to the efficient calculation of the local kinetic energy through a specifically designed lowrank update, which is crucial for preserving the cubic scaling of the ansatz. A complete documentation of the lowrank update is provided in appendix E. Benchmark results for a two-dimensional $t - V$ model of spinless fermions are shown in Sec. 3. We conclude with an outlook on possible extensions to autoregressive multireference wavefunctions in Sec. 4 and summarize in Sec. 5.

## 2 Method

We consider a model of $N_p$ spinless fermions on $N_s$ lattice sites with Hamiltonian

$$\mathcal{H} = \mathcal{H}_{\text{int}}(\{n\}) - \frac{1}{2} \sum_{r,s=1}^{N_s} t_{sr} \left( c_s^\dagger c_r + c_r^\dagger c_s \right). \tag{5}$$

The fermion operators satisfy canonical anticommutation rules $\{c_r^\dagger, c_s\} = \delta_{rs}$ and the occupation numbers are $n_r = c_r^\dagger c_r$. The interactions $\mathcal{H}_{\text{int}}$ are assumed to be diagonal in the basis of occupation number configurations $\{n\} = (n_1, \ldots, n_{N_s})$. Variational Monte Carlo (VMC) for simulating ground state physics is based on the Rayleigh-Ritz variational principle. With the definition of the local energy

$$E_\theta^{\text{loc}}(x) = \frac{\langle x | \mathcal{H} | \psi_\theta \rangle}{\langle x | \psi_\theta \rangle}, \tag{6}$$

for basis state $x \equiv \{n\}$ the expectation value of the energy over the variational wavefunction is

$$\begin{aligned} \langle E_\theta \rangle &= \sum_x \frac{|\langle x | \psi_\theta \rangle|^2}{\langle \psi_\theta | \psi_\theta \rangle} \frac{\langle x | \mathcal{H} | \psi_\theta \rangle}{\langle x | \psi_\theta \rangle} \\ &= \sum_x p_\theta(x) E_\theta^{\text{loc}}(x). \end{aligned} \tag{7}$$

$\langle E_\theta \rangle$ is minimized iteratively with respect to the variational parameters $\theta$ and needs to be estimated at each step. Since

$$p_\theta(x) = \frac{|\psi_\theta(x)|^2}{\sum_{x'} |\psi_\theta(x')|^2} \tag{8}$$

is positive definite, it can be interpreted as a probability distribution and the high-dimensional sum in Eq. 7 is evaluated using a Monte Carlo scheme

$$\langle E_\theta \rangle \approx \frac{1}{N_{\text{samples}}} \sum_{x \sim p_\theta(x)} E_\theta^{\text{loc}}(x), \tag{9}$$

where $x \sim p_\theta(x)$ denotes drawing a sample from the probability distribution $p_\theta(x)$. Note that amplitude and sign of the wavefunction $\psi_\theta(x) = \langle x | \psi_\theta \rangle$ appear only in the expression for the local energy

$$E_\theta^{\text{loc}}(x) = \sum_{x'} \frac{\langle x | \mathcal{H} | x' \rangle \langle x' | \psi_\theta \rangle}{\langle x | \psi_\theta \rangle} \tag{10}$$

through a ratio of amplitudes involving all basis states $x'$ connected to $x$ via the (off-diagonal part of the) Hamiltonian. An autoregressive variational ansatz must fulfill two tasks:

1. Direct sampling: Generate uncorrelated samples $x$ from the normalized probability distribution $p_\theta(x)$.

2. Probability density estimation: For calculating the local kinetic energy (or any other off-diagonal observable) the wavefunction amplitude $\langle x' | \psi_\theta \rangle = \text{sign}(\langle x' | \psi_\theta \rangle) \sqrt{|\langle x' | \psi_\theta \rangle|^2}$ on an arbitrary (i.e. not necessarily previously sampled) configuration $x'$ must be available to obtain the amplitude ratio $\langle x' | \psi_\theta \rangle / \langle x | \psi_\theta \rangle$ in Eq. (10).

Furthermore gradients with respect to the variational parameters $\theta$ need to be calculated, which is conveniently done using automatic differentiation.

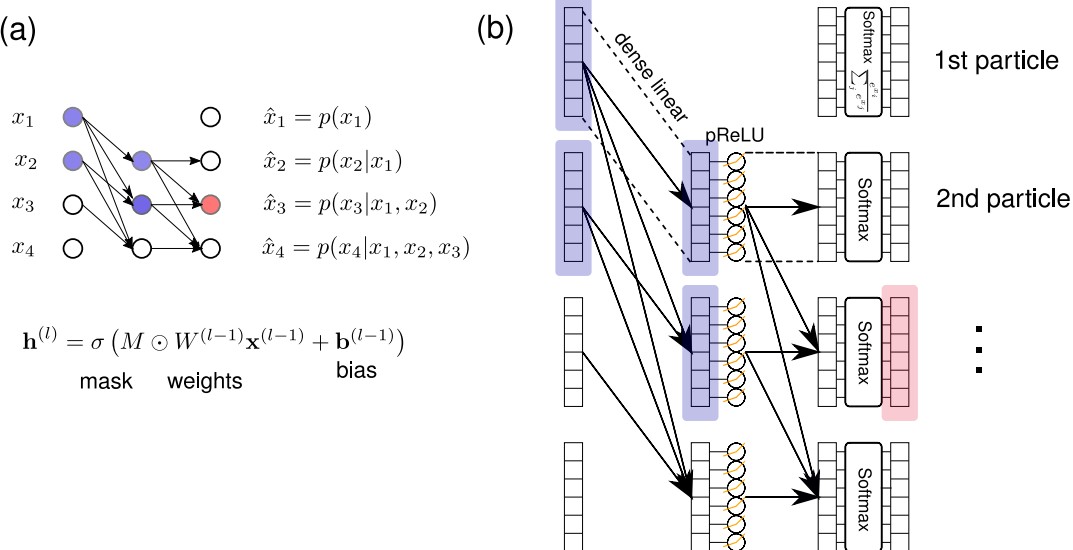

Figure 2: (a) Three-layer MADE network for binary components: By zeroing out elements of the network weight matrix $W$ with a mask $M$ of zeros and ones it is ensured that outputs $\hat{x}_{k'}$ with index $k'$ only depend on inputs $x_k$ with index $k < k'$. The activation $\sigma(a) = (1 + \exp(-a))^{-1}$ at each output ensures that the output $\hat{x}_k \in [0,1]$ can be interpreted as the probability for the binary variable $x_k$ to be set to 1, which is conditional on the inputs in the receptive field of $\hat{x}_k$. (b) MADE Jastrow network for sampling particle positions: Inputs are one-hot encoded particle positions, outputs are categorical distributions over the number of sites where the $k$-th particle is to be placed. Connections in the original MADE network (a) are replaced by dense linear layers between blocks of $N_s$ nodes each. The activation function at each node is a parametric ReLU [62].

## 2.1 Neural network Jastrow factor

### 2.1.1 MADE architecture

Many neural network architectures satisfying the autoregressive property have been put forward. We use the masked autoencoder for distribution estimation (MADE) proposed by Germain et al. [21], which makes both sampling and density estimation (inference) tractable. The autoregressive connectivity between input and output nodes is realized by appropriate masks in a fully-connected feed-forward neural network, see Fig. 2(a), and all conditional probabilities can be obtained in a single forward pass. Sampling requires $N$ forward passes, where $N$ is the number of "components" [21].

For modeling the Jastrow factor, the inputs are adapted to represent positions of indisinguishable particles rather than binary variables, see Fig. 2(b). A configuration in the computational basis of $N_p$ indistinguishable particles (spinless fermions) on $N_s$ sites is either specified in terms of ordered particle positions $|x\rangle \equiv |i_1, i_2, \ldots, i_{N_p}\rangle$ with $i_1 < i_2 < \ldots < i_{N_p}$ or, equivalently, in terms of occupation numbers $n_i \in \{0,1\}$, i.e. $|x\rangle = |n_1, n_2, \ldots, n_{N_s}\rangle$ where $n_i = 1$ if $i \in \{i_1, i_2, \ldots, i_{N_p}\}$ and $n_i = 0$ otherwise. Conceptually, the autoregressive Slater-Jastrow ansatz is a generative model for ordered particle *positions* at fixed total particle number, and it is autoregressive in the particle index (see Fig. 2(b)), however, as will be argued below, the representation in terms of occupation numbers is essential in an intermediate step (see Sec. 2.2.2).

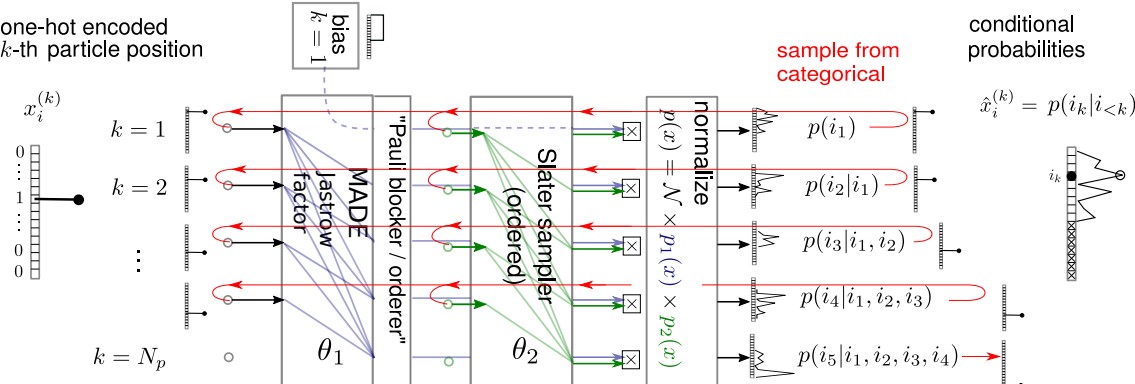

Figure 3: Architecture of the autoregressive Slater-Jastrow ansatz. Note that the Slater sampler and Jastrow network act in parallel and not sequentially. The last input is not connected to any output as this would violate the autoregressive property. The first output is unconditional and the probability is solely determined by the Slater determinant $p(i_1) = |\langle i_1 | \psi_{SD} \rangle|^2$. Red arrows indicate the iterative sampling along the chain of conditional probabilities in which a sampled on-hot encoded particle position is fed back to the inputs such that another pass through the network determines the next conditional probability. The probability of the generated sample is the product of the conditional probabilities at the actually sampled positions.

A configuration $|i_1, i_2, \ldots, i_{N_p}\rangle$ is "unfolded" and input to the autoregressive Slater-Jastrow network (Fig. 3) as a concatenation of $N_p$ blocks of one-hot encoded particle positions

$$x^{\text{in}} = [\text{onehot}(i_1), \text{onehot}(i_2), \ldots, \text{onehot}(i_{N_p})],$$

which componentwise reads $x^{\text{in}}_{(k-1)N_s+1:kN_s} = \delta_{i,i_k}$ with $i, i_k \in \{1, \ldots, N_s\}$ and $k = 1, \ldots, N_p$. Here, $[\cdot, \cdot]$ denotes vector concatenation and the colon notation

$$i : j = (i, i+1, i+2, \ldots, j)$$

indicates a range of indices. The output of the autoregressive Slater-Jastrow network

$$\hat{x}^{\text{out}} = [p(i_1), p(i_2|i_1), p(i_3|i_1, i_2), \ldots, p(i_{N_p}|i_{<N_p})]$$

is a concatenation of $N_p$ blocks of categorical distributions over the number of sites with the $k$-th block $p(i_k|i_{<k}) \equiv p_{\text{cond}}(k, i_k)$ modeling the conditional probability for the $k$-th particle to be at position $i_k$. The normalization $\sum_{i_k=1}^{N_s} p(k, i_k) = 1$ of each conditional probability is ensured by applying a softmax activation function in the last layer to each block of outputs.

The total input dimension of the MADE network for particle positions is $N_p N_s$, as there are $N_p$ blocks of one-hot encodings of $N_s$ particle positions each. The weights connecting nodes in the original MADE architecture [20, 21] are promoted to $N_s \times N_s$ weight matrix blocks. Thus the dimensionality of the weight matrices between layers is $N_p N_s \times N_p N_s$, and to ensure the autoregressive property they have a lower-triangular block structure with blocks of size

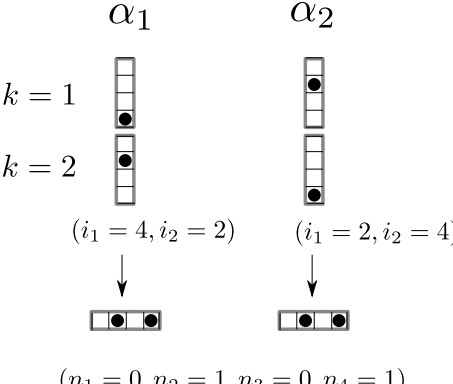

Figure 4: Distinct "unfolded" output configurations of the autoregressive Jastrow neural network correspond to the same configuration of indistinguishable particles. In $\alpha_1$ the "first" particle is at position 4 and the "second" particle at position 2, while in $\alpha_2$ the order is reversed. $\alpha_1$ and $\alpha_2$ represent the same state of indistinguishable particles, but for the network they appear distinct and are assigned different probabilities.

$N_s \times N_s$. Since direct connections are allowed for all layers except the first one (so as to ensure the autoregressice property [21]) only the weight matrix of the first layer is strictly lower block-triangular, whereas the weight matrices of subsequent layers can have non-zero blocks on the diagonal.

### 2.1.2 Permutation invariant autoregressive Jastrow factor

The autoregressive ansatz implies some predetermined ordering of inputs and outputs. If the outputs are interpreted as categorial distributions for particle configurations without any constraints, MADE will sample distinct unfolded configurations which correspond to the same state of indistinguishable particles. However, they would be assigned different probabilities by the Jastrow network (see Fig. 4). Therefore it must be ensured that only one of the possible $N_p!$ permutations of $N_p$ particle positions is output by the network so that a unique probability can be assigned to each configuration of indistinguishable particles. This is achieved by imposing an ordering constraint [63] on the particle positions:

$$i_1 < i_2 < \ldots < i_{N_p}.$$

It is implemented by augmenting the ouput blocks of MADE for each particle $k$ by a "block-ing"/"ordering" layer which modifies the output such that the conditional probability for the $k$-th particle to sit at site $i$ is

$$\hat{x}_i(k) = \begin{cases} 0, & \text{if } i \leq i_{k-1}, \\ 0, & \text{if } i \geq N_s - (N_p - k) + 1, \\ \hat{x}_i(k)/(\sum_{j=i_{k-1}+1}^{N_s-(N_p-k)} \hat{x}_j(k)), & \text{else}. \end{cases} \tag{11}$$

The first line in Eq. (11) expresses the requirement that the $k$-th particle has to be "to the right" of $(k-1)$-th while the second line is due to the fact that the remaining $(N_p - k)$ particles need space somewhere "to the right" (Pauli blocking). The third line returns a normalized probability on the support of valid positions for the $k$-th particle, which is $i_{\min}[k] \leq i_k \leq i_{\max}[k]$ with $i_{\min}[k] = i_{k-1} + 1$ and $i_{\max}[k] = N_s - (N_p - k) - 1$. The Jastrow MADE network fulfills two computational tasks:

1. Sampling: During the sampling, which requires $N_p$ passes through the MADE network [21], automatic differentiation is not activated. The MADE network can process a batch of samples in parallel, however note that the final "Pauli ordering" layer and the normalization in Eq. (11) depend on the given particle configuration. The position of the first particle (which is unconditional) is sampled from the probability distribution given in terms of the Slater determinant only (see following section, as well as Fig. 5). To this end the first output block of the "Pauli ordering" layer is set to a uniform distribution.

2. For density estimation only a single pass through MADE in necessary [21].

The MADE network is implemented in the PyTorch [64] machine learning framework.

## 2.2 Slater sampler

In statistical analysis the problem of directly sampling fermion configurations from a Slater determinant is known as a "determinantal point process" [54, 55] and fast algorithms scaling as $\mathcal{O}(N_p N_s^2)$ have been designed [54, 65, 66]. The following Sec. 2.2.1 re-derives such a fast fermion sampling algorithm, where particle positions that are successively sampled come out *"unordered"*. Although this is not the actual fermion sampling algorithm that will be used, it introduces the optimizations that are crucial for achieving cubic scaling with system size and sets the stage for the efficient *"ordered"* sampling algorithm in Sec. 2.2.2, which is the one relevant to the autoregressive Slater-Jastrow ansatz. The reader interested only in the final algorithm may directly proceed to Sec. 2.2.2.

As a matter of terminology, the "unordered" sampling is in fact based on the first-quantized formalism, where the particles are regarded as distinct and the anti-symmetrization of the wave function is added as a requirement. By contrast, the "ordered" sampling is based on the second-quantized formalism, where we no longer distinguish particle labels.

### 2.2.1 First-quantized ("unordered") direct sampling

Autoregressive sampling is feasible whenever the probability distribution is tractable, i.e. its normalization constant and the marginal distributions over components can be calculated efficiently. For a Slater determinant this is possible. The Hartree-Fock Slater determinant is written in terms of the matrix $P$ of single-particle orbitals as

$$|\psi_0\rangle = \prod_{n=1}^{N_p} \sum_{i=1}^{N_s} P_{i,n} c_i^\dagger |0\rangle .$$

The corresponding one-body density matrix is denoted as $\{\langle\psi_0|c_i^\dagger c_j|\psi_0\rangle\}_{i,j=1}^{N_s} = PP^\dagger \equiv \mathcal{G}$ and the single-particle Green's function as $G = \mathbb{1}_{N_s} - \mathcal{G}$. The marginal probability for $k \leq N_p$ particles to be at positions $(i_1, i_2, \ldots, i_k)$ is given by the generalized Wick's theorem in terms of principal minors of the one-body density matrix

$$p(i_1, i_2, \ldots, i_k) = \sum_{i_{k+1}, \ldots, i_{N_p}=1}^{N_s} |\langle i_1, i_2, \ldots, i_k, i_{k+1}, \ldots, i_{N_p}|\psi_0\rangle|^2$$

$$= \frac{1}{k!\binom{N_p}{k}} \det\left(\mathcal{G}_{I_k, I_k}\right) . \tag{12}$$

$\mathcal{G}_{I_k, I_k}$ is the restriction of the matrix $\mathcal{G}$ to the rows and columns given by the ordered index set of particle positions $I_k = \{i_1, i_2, \ldots, i_{k-1}, i_k\}$. In this section the index $k$ labelling position

$i_k$ denotes the sampling step and, other than that, there is no ordering among the positions implied. The conditional probability can be obtained from the ratio of marginals

$$p(i_{k+1}|i_1,\ldots,i_k) = \frac{p(i_1,i_2,\ldots,i_{k+1})}{p(i_1,i_2,\ldots,i_k)}\,. \tag{13}$$

Note that the normalization constant in Eq. (12) does not depend on the sample $(i_1,i_2,\ldots,i_k)$. This normalization is only valid if the one-body density matrix can be written as $\mathcal{G} = PP^\dagger$, i.e. it derives from a single Slater determinant and therefore is a projector, $\mathcal{G}^2 = \mathcal{G}$. The normalization will be dropped in the following and restored in the final result. The normalized conditional probabilities are

$$p(i_{k+1}|i_{<k+1}) = \frac{1}{N-k}\tilde{p}(i_{k+1}|i_{<k+1})\,, \tag{14}$$

where we denote by a tilde the *unnormalized* conditional probabilities

$$
\begin{aligned}
\tilde{p}(i_{k+1}|i_{<k+1}) &= \frac{\det\left(\mathcal{G}_{I_{k+1},I_{k+1}}\right)}{\det\left(\mathcal{G}_{I_k,I_k}\right)} \\
&= \frac{\det\begin{pmatrix} X[k] & \mathcal{G}_{I_k,i_{k+1}} \\ \mathcal{G}_{I_k,i_{k+1}}^T & \mathcal{G}_{i_{k+1},i_{k+1}} \end{pmatrix}}{\det(X[k])} \\
&= \mathcal{G}_{i_{k+1},i_{k+1}} - \sum_{l,m\in I_k} \mathcal{G}_{i_{k+1},l}\left(X[k]^{-1}\right)_{l,m}\mathcal{G}_{m,i_{k+1}}\,.
\end{aligned}
\begin{matrix} \\ \tag{15} \\ \tag{16} \end{matrix}
$$

$$\tag{17}$$

Here, the $k \times k$ submatrix $\mathcal{G}_{I_k,I_k}$ is abbreviated as $X[k]$. The block determinant formula has been used, which allows to cancel the denominator determinant $\det(X[k])$ so that the determinant of the Schur complement of $X[k]$ remains. Whenever numerator and denominator matrix differ in just one row and column, the latter is just a number. The expression for $\tilde{p}(i_{k+2}|i_{<k+2})$ is analogous, with $X[k]$ replaced by $X[k+1]$. Having $X[k]^{-1}$ available is essential for efficient calculation of the Schur complement for all positions $i_{k+1} \in \{i_1,\ldots,i_{N_s}\}$. While traversing the chain of conditional probabilities and sampling new particle positions conditional on the previously sampled ones, we need to keep track of the inverse matrices $X^{-1}[1] \to X^{-1}[2] \to \ldots X^{-1}[k-1] \to X^{-1}[k]$ and update $X^{-1}[k]$ iteratively based on $X^{-1}[k-1]$ using the block matrix update formula. With the definition of the vector

$$\vec{\xi}[k-1] = X^{-1}[k-1]\,\mathcal{G}_{I_{k-1},i_k}\,, \tag{18}$$

the block update of the inverse matrix is seen to be a low-rank update

$$
\begin{aligned}
X^{-1}[k] &= \begin{pmatrix} X[k-1] & \mathcal{G}_{I_{k-1},i_k} \\ \mathcal{G}_{I_{k-1},i_k}^T & \mathcal{G}_{i_k,i_k} \end{pmatrix}^{-1} \\
&= \left(\begin{array}{c|c} X^{-1}[k-1] + \vec{\xi}[k-1]\otimes S^{-1}[k]\otimes\vec{\xi}[k-1]^T & -\vec{\xi}[k-1]S^{-1}[k] \\ \hline -S^{-1}[k]\vec{\xi}[k-1]^T & S^{-1}[k] \end{array}\right).
\end{aligned} \tag{19}
$$

$S^{-1}[k] \equiv \tilde{p}(i_k|i_{<k}[\text{unordered}])$, the inverse of the Schur complement of $X[k-1]$, is given by Eq. (16) evaluated at the *actually sampled position* $i_k$. Therefore $S^{-1}[k]$ has already been computed, and in the given case of unordered sampling it is just a scalar (compare with Sec. 2.2.2, where the Schur complement is a matrix). In terms of computational complexity, the matrix-vector product in Eq. (18) costs $\mathcal{O}((k-1)^2)$ while the expression in the first block of Eq. (19)

is an outer product, which costs $\mathcal{O}((k-1)^2)$. Thus, after sampling the position of the $k$-th particle, the update $X^{-1}[k-1] \to X^{-1}[k]$ can be preformed with $\mathcal{O}((k-1)^2)$ operations.

The matrix-vector product in Eq. (16) also costs $\mathcal{O}(k^2)$, but it needs to be computed for the conditional probabilities at all $i_{k+1} \in \{1, \ldots, N_s\}$, resulting in a cost of $\mathcal{O}(N_s k^2)$ at the $(k+1)$-th sampling step. Then sampling all $N_p$ particle positions $\{i_1, i_2, \ldots, i_{N_p}\}$ along the entire chain of conditional probabilities would cost $\mathcal{O}(N_s N_p^3)$, if Eq. (16) were used directly.

Making use of computations done at previous iterations [65, 66], a recursion relation for the conditional probabilities can be derived. By inserting $X^{-1}[k]$ from Eq. (19) into Eq. (16) and recognizing the expression for the conditional probabilities at the previous sampling step $k$, it can be verified that

$$\tilde{p}(i_{k+1} = x | i_{<k+1}) = \tilde{p}(i_k = x | i_{<k}) - S^{-1}[k]\chi(x)^2, \tag{20}$$

where

$$\chi(x) = \left( \sum_{m \in I_{k-1}} \xi[k-1]_m \mathcal{G}_{m,x} \right) - \mathcal{G}_{i_k, x}, \tag{21}$$

and, as stated earlier, $S^{-1}[k] = \tilde{p}(i_k | i_{<k})$ is the *unnormalized* conditional probability for the position of the $k$-th particle at the actually sampled position $i_k$. In effect, the matrix-vector product in Eq. (16) has been replaced by a vector-vector dot product in Eq. (21), which reduces the computational cost at sampling step $k$ to $\mathcal{O}(N_s k)$ and the cost of sampling all $N_p$ particle positions $(i_1, i_2, \ldots, i_{N_p})$, including the iterative update of $X^{-1}[k]$, to $\sum_{k=1}^{N_p} (k^2 + N_s k) \sim \mathcal{O}(N_s N_p^2)$ for $N_s > N_p$.

The presented algorithm is similar to "Algorithm 2" in Ref. [65] and "Algorithm 3" in Ref. [66], except that there the explicit construction of the matrix $X^{-1}[k]$ has also been avoided. Note that another fast fermion sampling algorithm scaling as $\mathcal{O}(N_s N_p^2)$ is given in Ref. [54].

### 2.2.2 Second-quantized ("ordered") direct sampling

A Slater determinant is by construction invariant under permutation of particle positions, i.e.

$$|\langle \sigma(i_1)\sigma(i_2) \ldots \sigma(i_{N_p}) | \psi_0 \rangle|^2 = |\langle i_1 i_2 \ldots i_{N_p} | \psi_0 \rangle|^2, \tag{22}$$

where $\sigma$ is an element of the symmetric group $\mathcal{S}_{N_p}$ of $N_p$ permutations. This is reflected in Eq. (12) by the fact that an equal number of row and column permutations does not change the determinant. As mentioned earlier, the same is not true for the autoregressive Jastrow factor, and one needs to impose an ordering constraint to be able to assign unique probabilities to configurations of indistinguishable particles. Now, the statement that the second particle is at position $i_2$ and is "to the right" in the chosen fermion ordering of the first particle at position $i_1$, that is $i_2 > i_1$, actually implies that all positions between $i_1$ and $i_2$ are empty. This cannot be guaranteed by first-quantized ("unordered") sampling from a Slater determinant, which is therefore incompatible with the ansatz for the autoregressive Jastrow factor. Instead, one needs to sample sequentially (for example in a snake-like ordering in dimension $D \geq 2$, see Fig. 1) occupation numbers rather than particle positions to make sure that the sites between $i_k$ and $i_{k-1}$ are empty and the particle position sampled in the $k$-th sampling step is also the $k$-th one in the fermion ordering. This is outlined in the following.

The joint (marginal) distribution of a subset of occupation numbers is [27]

$$p(n_1, n_2, \ldots, n_m) = (-1)^{\sum_{i=1}^{m} n_i} \det \begin{pmatrix} G_{1,1} - n_1 & G_{1,2} & \cdots & G_{1,m} \\ G_{2,1} & G_{2,2} - n_2 & \cdots & G_{2,m} \\ \vdots & \vdots & \ddots & \vdots \\ G_{m,1} & G_{m,2} & \cdots & G_{m,m} - n_m \end{pmatrix}, \tag{23}$$

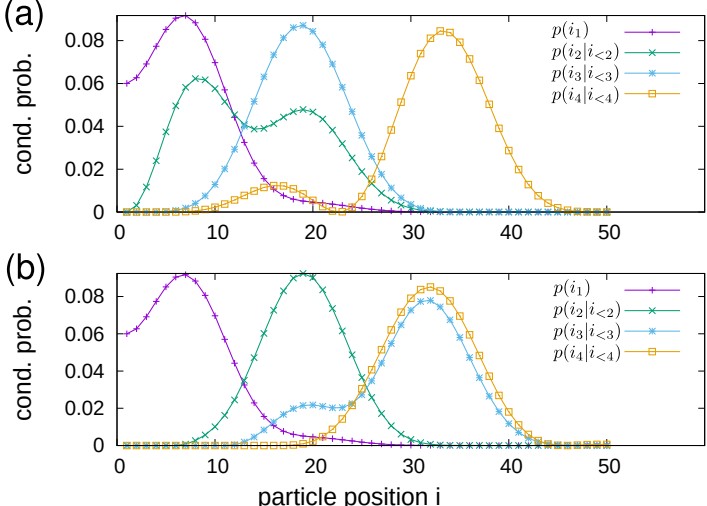

Figure 5: Conditional probabilities for a Slater determinant of $N_p = 4$ non-interacting fermions in a chain with $N_s = 50$ sites and periodic boundary conditions. For the configuration in (a) the first particle is at position $i_1 = 1$ and the second and third at $i_2 = 2, i_3 = 3$. In (b) the positions of the first three particles are $i_1 = 5, i_2 = 10$ and $i_3 = 15$. Clearly, there are conditional probabilities which approach zero due to interference (not caused by the Pauli principle). Note that the probability for the first particle, which is unconditional, is not uniform because of the requirement that all positions to the left be empty.

where $G_{i,j}$ are elements of the single-particle Green's function. Note that $p(n_1, n_2, \ldots, n_m)$ in Eq. (23) is correctly normalized. In terms of the joint distribution of occupation numbers the joint distribution of ordered particle positions can be expressed as

$$p(i_1 < i_2 < \ldots < i_k = m) = p(n_1 = 0, n_2 = 0, \ldots, n_{i_1} = 1, \ldots,$$

$$n_{i_2 - 1} = 0, n_{i_2} = 1, n_{i_2 + 1} = 0, \ldots, n_m = 1). \tag{24}$$

With the obvious convention that occupation numbers at particle positions are equal to one and between particle positions equal to zero, the conditional probability $p(i_{k+1} | i_{<k+1}[\text{ordered}])$ for ordered particle positions is

$$p(i_{k+1} | i_{<k+1}[\text{ordered}]) = \frac{p(i_1 < i_2 < \ldots < i_k = m < i_{k+1} = m + l)}{p(i_1 < i_2 < \ldots < i_k = m)} \tag{25}$$

$$= (-1)^1 \frac{\det \begin{pmatrix} G_{1,1} - n_1 & G_{1,2} & \cdots & G_{1,m} & G_{1,m+1} & \cdots & G_{1,m+l} \\ G_{2,1} & G_{2,2} - n_2 & \cdots & G_{2,m} & G_{2,m+1} & \cdots & G_{2,m+l} \\ \vdots & \vdots & \ddots & \vdots & \vdots & \vdots & \vdots \\ G_{m,1} & G_{m,2} & \cdots & G_{m,m} - n_m & G_{m,m+1} & \cdots & G_{m,m+l} \\ G_{m+1,1} & G_{m+1,2} & \cdots & G_{m+1,m} & G_{m+1,m+1} & \cdots & G_{m+1,m+l} \\ \vdots & \vdots & \vdots & \vdots & \vdots & \ddots & \vdots \\ G_{m+l,1} & G_{m+l,2} & \cdots & G_{m+l,m} & G_{m+l,m+1} & \cdots & G_{m+l,m+l} + 1 \end{pmatrix}}{\det \begin{pmatrix} G_{1,1} - n_1 & G_{1,2} & \cdots & G_{1,m} \\ G_{2,1} & G_{2,2} - n_2 & \cdots & G_{2,m} \\ \vdots & \vdots & \ddots & \vdots \\ G_{m,1} & G_{m,2} & \cdots & G_{m,m} - n_m \end{pmatrix}}, \tag{26}$$

where $i_k \equiv m$ and $i_{k+1} \equiv m + l$ are the positions of the $k$-th and $(k+1)$-th particle in the given fermion ordering. Like Eq. (15), the numerator matrix exhibits a block structure of the form

$$(26) \equiv \left( \begin{array}{c|c} \tilde{X}[k] & B \\ \hline B^T & D \end{array} \right), \tag{27}$$

where $\tilde{X}[k]$ is an $i_k \times i_k$ matrix and the blocks $B$ and $D$ are defined by Eq. (26). Using again the block determinant formula and cancelling $\det(\tilde{X}[k])$, we are left with the determinant of the Schur complement of $\tilde{X}[k]$:

$$p(i_{k+1}|i_{<k+1}[\text{ordered}]) = (-1)\det\left(D - B^T \tilde{X}^{-1}[k]B\right). \tag{28}$$

At variance with Eq. (15), where the blocks $B$ and $D$ are a vector and a number, respectively, in Eq. (26) the width $l$ of those blocks is in the range $l \in \{1, \ldots, i_{\max}[k+1] - i_k\}$.

The inverse matrix $\tilde{X}^{-1}[k+1]$ is updated iteratively from $\tilde{X}^{-1}$ to be available when calculating the Schur complement in the next sampling step $k+1$. As in Eq. (19) the formula for the inverse of a block matrix is used:

$$\tilde{X}^{-1}[k+1] = \left( \begin{array}{cc} \tilde{X}[k] & B \\ B^T & D \end{array} \right)^{-1}. \tag{29}$$

For ordered sampling the Schur complement of $\tilde{X}[k]$,

$$\tilde{S}[k+1] \equiv D - B^T \tilde{X}^{-1}[k]B \tag{30}$$

is an $l \times l$ matrix. Direct calculation according to Eqs. (29) and (30) costs $\mathcal{O}(i_k^2 l + l^2 i_k)$ operations, and at the $(k+1)$-th sampling step all conditional probabilities on the allowed support $l \in \{1, \ldots, i_{\max}[k+1] - i_k\}$, see Eq. 11, need to be computed.

By reusing previously calculated matrix-vector products in the Schur complement in Eqs. (29) and (30) for different values of $l$ (see appendix D) as well as by reusing computations already done when calculating the conditional probabilities of the previous particle (similar to the algorithm in Sec. 2.2.1), the overall computation cost for producing an uncorrelated sample of *ordered* particle positions can be brought down to $\mathcal{O}(N_s^3)$, see Fig. 6. A precise operation count is difficult due to the stochastic dependence on the sampled positions $\{i_k\}$. For small system sizes the computational cost is dominated by linear algebra operations for very small matrices [67]. The speedup related to optimized ordered sampling (see appendix D) only comes to bear when the system size is large enough so that reusing precomputed matrix-vector products pays off.

As the ordered sampling proceeds by calculating conditional probabilities for occupation numbers rather than particle positions, the sampling space is initially the grand-canonical ensemble which is then by the trick (11) constrained to fixed particle number. The computational complexity is therefore only weakly dependent on the particle number or filling (see Fig. 7).

### 2.2.3 Exploiting normalization

The fact that the conditional probabilities for each component of the Slater determinant sampler are normalized can be exploited to "screen" the probabilities: The conditional probability $p_{\text{cond}}(k, i_k)$ for the $k$-th component is calculated consecutively for the orbitals $i_k = i_{\min}, \ldots, m, \ldots, i_{\max}$ up to the smallest $m$ for which the normalization of probabilities $\sum_{i_k=1}^{m} p_{\text{cond}}(k, i_k) > 1 - \varepsilon$ is exhausted within a small margin $\varepsilon$. Calculations for $i > m$ are skipped and the corresponding probabilities are set to zero. It is found that with $\varepsilon = 10^{-10} - 10^{-8}$, approximately 25% of the conditional probabilities that would need to be evaluated can be skipped without affecting the normalization. Appendix F shows what conditional probabilities in a large two-dimensional system of non-interacting fermions look like.

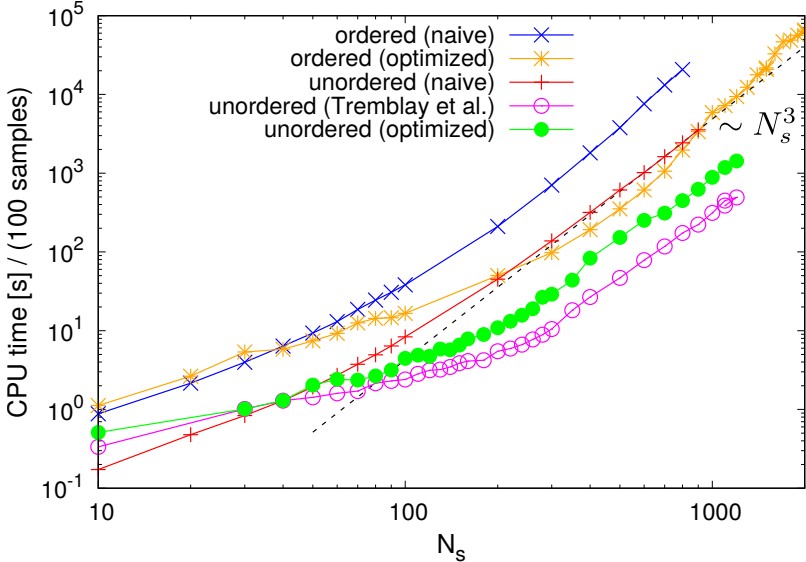

Figure 6: CPU time in seconds for generating 100 samples from a Slater determinant as a function of system size $N_s$ for half filling, $N_p = N_s/2$. Both unordered and ordered direct sampling of particle positions scales like $\mathcal{O}(N_s^3)$, which is indicated by the dashed line. Shown for *unordered* direct sampling are the naive algorithm of Sec. 2.2.1 without reusing previous computations (red), the optimized algorithm of Sec. 2.2.1 (green), and the algorithm of Tremblay at al. [66] (magenta). Shown for *ordered* direct sampling is the naive implementation in Sec. 2.2.2 without reusing previous computations (blue) and the optimized version described in appendix D (orange).

## 2.3 Normalization of MADE × SD

Having discussed the direct sampling of positions of indistinguishable particles from the symmetric Jastrow factor (MADE) and the anti-symmetric Slater determinant sampler (SD), we now turn to the coupling of the two autoregressive generative models. In machine learning terminology such multiplication of two model probabilities is known as a *product of experts* [68]. This brings about the issue of normalization since the product of two individually normalized probability distributions, $\sum_{\{\mathbf{n}\}} p_{\text{SD}}(\mathbf{n}) = 1$ and $\sum_{\{\mathbf{n}\}} p_{\text{Jastrow}}(\mathbf{n}) = 1$, is not normalized.

Due to the structure of the autoregressice ansatz, the normalization is done at the level of the conditional probabilities, i.e. for each output block of MADE × SD, which is feasible since the size of their support is at most $N_s - N_p + 1$. The normalized modulus squared of the wavefunction for a configuration $|\beta\rangle$ with occupation numbers $\mathbf{n}^{(\beta)}$ in the combined autoregressive Slater-Jastrow ansatz reads

$$p_\theta(\mathbf{n}^{(\beta)}) \equiv |\langle \beta | \Psi_\theta \rangle|^2 = \frac{p_{\text{Jastrow}}(\mathbf{n}^{(\beta)}) \times p_{\text{SD}}(\mathbf{n}^{(\beta)})}{\mathcal{N}(\mathbf{n}^{(\beta)})}, \tag{31}$$

where

$$p_{\text{SD}}(\mathbf{n}^{(\beta)}) = \prod_{k=1}^{N_p} p_{\text{SD}}(n_{i_k}^{(\beta)} | n_{i<i_k}^{(\beta)}), \tag{32}$$

and

$$p_{\text{Jastrow}}(\mathbf{n}^{(\beta)}) = \prod_{k=1}^{N_p} p_{\text{MADE}}(n_{i_k}^{(\beta)} | n_{i<i_k}^{(\beta)}) \tag{33}$$

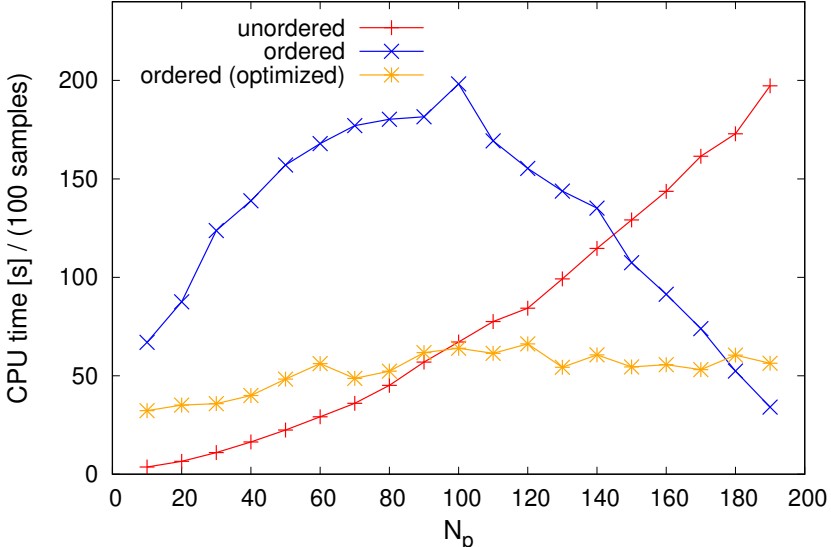

Figure 7: CPU time in seconds for generating 100 samples from a Slater determinant as a function of the number of particles $N_p$ on $N_s = 200$ sites, using ordered and unordered sampling. For ordered sampling the computational effort decreases again for particle numbers above half filling because the number of available positions is restricted by the Pauli blocker.

is the (normalized) probability in the Slater determinant or Jastrow ansatz, written as a chain of conditional probabilities. It is easy to show that the normalization of all conditional probabilities implies the correct normalization of the joint distribution. Pairing up corresponding conditional probabilities in Eqs. (32) and (33) and normalizing over the relevant support leads to

$$p_{\text{SJ}}(n_k(i_k) = 1 | n_1, n_2, \ldots, n_{k-1})$$
$$= \frac{p_{\text{SD}}(n_k(i_k) = 1 | n_1, \ldots, n_{k-1}) \cdot p_{\text{MADE}}(n_k(i_k) = 1 | n_1, \ldots, n_{k-1})}{\sum_{i_k \in I_k} p_{\text{SD}}(n_k(i_k) = 1 | n_1, \ldots, n_{k-1}) p_{\text{MADE}}(n_k(i_k) = 1 | n_1, \ldots, n_{k-1})}. \tag{34}$$

Here, the sum runs over the support for the $k$-th sampling step, which according to Eq. (11) is $I_k = [i_{\min}, i_{\max}] \equiv [i_{k-1} + 1, N_s - (N_p - k)]$. It should be emphasized that $p_{\text{SJ}}(n_k(i_k) = 1 | n_1, \ldots, n_{k-1})$ means the conditional probability that the $k$-th particle sits at position $i_k$ with all positions between $i_{k-1}$ and $i_k$ empty. All normalization constants can be grouped together into

$$\mathcal{N}(\mathbf{n}^{(\beta)}) = \left( \sum_{i_1 \in I_1} p_{\text{SD}}(n_1(i_1) = 1) \cdot p_{\text{MADE}}(n_1(i_1) = 1) \right)$$
$$\cdot \left( \sum_{i_2 \in I_2} p_{\text{SD}}(n_2(i_2) = 1 | n_1) \cdot p_{\text{MADE}}(n_2(i_2) = 1 | n_1) \right) \cdots$$
$$\cdot \left( \sum_{i_{N_p} \in I_{N_p}} p_{\text{SD}}(n_{N_p}(i_{N_p}) = 1 | n_1, \ldots, n_{N_p-1}) \cdot p_{\text{MADE}}(n_{N_p}(i_{N_p}) = 1 | n_1, \ldots, n_{N_p-1}) \right), \tag{35}$$

which is the renormalization factor in Eq. (31). Of course, the conditional probabilities are sample-dependent, which is why also the renormalization factor $\mathcal{N}(\mathbf{n}^{(\beta)})$ depends on the given sample $\mathbf{n}^{(\beta)}$.

The normalization requirement has important implications for the inference step: Considering the Slater determinant and Jastrow network separately, the probability of a given sample $|\beta\rangle = |i_1, i_2, \ldots, i_{N_p}\rangle$ could be obtained from Eq. (12) for the Slater determinant, and for the Jastrow factor by passing the sample through the MADE network [21] and picking from the output of MADE only the conditional probabilities at the actually sampled positions and taking their product (according to Eq. (1)). On the other hand, in the combined model MADE × SD, for the sake of normalization according to Eq. (35), the conditional probabilities at *all* positions $i_k \in I_k$, not just at those of the given sample, need to be calculated even in the inference step. While MADE provides all conditional probabilities with a single pass through the network [21], the Slater sampler needs to traverse the full chain of sampling steps to generate all conditional probabilities since it has an internal state which needs to be updated iteratively. As a result, for the combined model, inference is as costly as sampling.

## 2.4 Sign of the wavefunction

The MADE network and the Slater sampler parameterize the probability, but not the amplitude and sign of a state $\psi_\theta(x) = \text{sgn}(\langle x|\psi_\theta\rangle)\sqrt{p_\theta(x)}$. The sign structure of the wavefunction is solely determined by the Slater determinant (which may be cooptimized with the Jastrow factor, i.e. rotated by an orthogonal matrix $R$, see Appendix C). Therefore

$$\text{sign}\left[\langle x|\psi_\theta\rangle\right] = \text{sign}\left[\langle x|\psi_0\rangle\right] \tag{36}$$

$$= \text{sign}\left[\det\left(P(R)_{\{i_1,i_2,\ldots,i_{N_p}\};\{1,2,\ldots,N_p\}}\right)\right], \tag{37}$$

which requires calculating the determinant of an $N_p \times N_p$ submatrix of the P-matrix in Eq. (2.2.1) and costs $\mathcal{O}(N_p^3)$ for each sample $x$. In the autoregressive approach subsequent samples $x$ are not related in any way and the sign needs to re-calculated for every sample. This does not modify the overall cubic scaling of our algorithm.

In the calculation of the local energy Eq. (10) the relative $\text{sign}(\langle\beta|\psi_0\rangle/\langle\alpha|\psi_0\rangle)$ is needed between a reference state $|\alpha\rangle$ and another basis state $|\beta\rangle$ differing just in the position of one particle. Given the local one-body density matrix

$$\mathcal{G}_{j,i} \equiv \mathcal{G}_{j,i}^{(\alpha,\psi_0)} = \frac{\langle\alpha|c_j^\dagger c_i|\psi_0\rangle}{\langle\alpha|\psi_0\rangle}, \tag{38}$$

it can be shown that the overlap ratio between $|\alpha\rangle$ and an occupation number state $|\beta\rangle$ differing from $|\alpha\rangle$ by a particle hopping from (occupied) site $r$ to (unoccupied) site $s$ is (see Appendix B)

$$\frac{\langle\beta|\psi_0\rangle}{\langle\alpha|\psi_0\rangle} = \left(1 - \mathcal{G}_{r,r} - \mathcal{G}_{s,s} + \mathcal{G}_{r,s} + \mathcal{G}_{s,r}\right) \times \sigma(r,s). \tag{39}$$

The additional sign

$$\sigma(r,s) = \langle\alpha|(-1)^{\sum_{i=\min(r,s)+1}^{\max(r,s)-1}\hat{n}_i}|\alpha\rangle \tag{40}$$

is due to the fact that in the P-matrix representation (2.2.1) of a Fock state columns need to be ordered according to increasing row index of particle positions. The index $i$ in Eq. (40) runs according to the fermion ordering. The calculation of the local OBDM (see Appendix B) requires the inversion of an $N_p \times N_p$ matrix and thus scales as $\mathcal{O}(N_p^3)$ with particle number $N_p$.

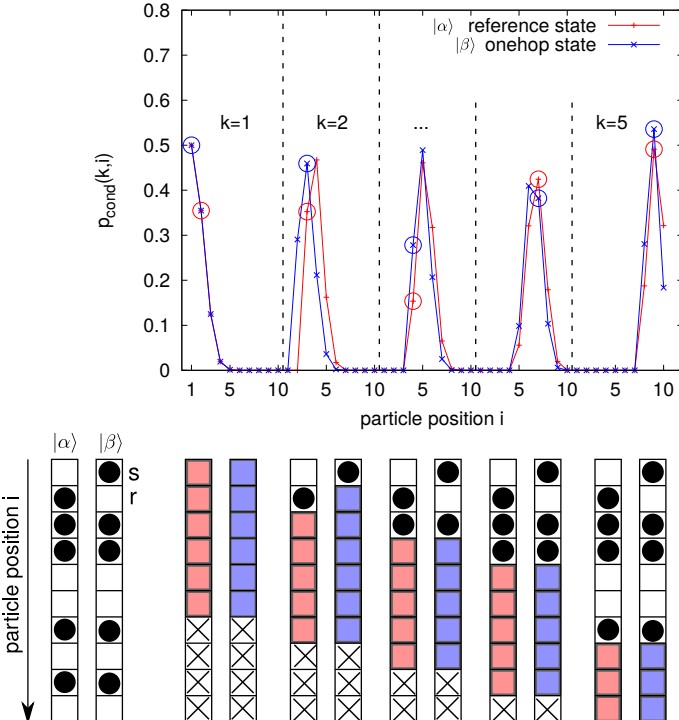

Figure 8: Example of conditional probabilities for a reference state $|\alpha\rangle$ and a corresponding one-hop state $|\beta\rangle$ for 5 particles on 10 sites. Open circles in the upper graph denote the conditional probabilities at the actual particle positions in state $|\alpha\rangle$ and $|\beta\rangle$, respectively. The conditional probabilities at other positions need to be calculated for the sake of obtaining the normalization constant in Eq. (35) (after multiplying with the probabilities of the Jastrow factor). Coloured shading in the lower part of the figure indicates the support of the distribution of conditional probabilities for sampling the position of the $k$-th particle in state $|\alpha\rangle$ (left set of sites, red) or $|\beta\rangle$ (right set of sites, blue), respectively.

## 2.5 Calculation of the local kinetic energy

### 2.5.1 The problem

The local energy of basis state $|\alpha\rangle$ is

$$E_\theta^{(\text{loc})}(\alpha) = \langle\alpha|H_{\text{int}}|\alpha\rangle + \sum_\beta \langle\alpha|H_{\text{kin}}|\beta\rangle \frac{\langle\beta|\psi_\theta\rangle}{\langle\alpha|\psi_\theta\rangle}, \tag{41}$$

where the non-zero contribution to the sum is from all states $|\beta\rangle$ connected to $|\alpha\rangle$ by single-particle hopping. Assuming that it exists, the occupation number state $|\beta\rangle \sim c_s^\dagger c_r |\alpha\rangle$ differs from $|\alpha\rangle$ only in the occupancies $n_r^{(\beta)} = 1 - n_r^{(\alpha)}$ and $n_s = 1 - n_s^{(\alpha)}$. The sign of $|\beta\rangle$ was discussed in the previous section. The configuration $|\beta\rangle$ is called a "one-hop state" as it arises from the "reference" state $|\alpha\rangle$ by the hopping of a single particle from position $r$ to $s$[1] In conventional Markov chain VMC the ratio of determinants $\langle\beta|\psi_0\rangle/\langle\alpha|\psi_0\rangle$ can be calculated using the lowrank update Eq. (39) so that $\langle\beta|\psi_0\rangle$ does not need to be calculated from scratch. Likewise, the ratio of Jastrow factors $J(\mathbf{n}^{(\beta)})/J(\mathbf{n}^{(\alpha)})$, which is diagonal in the occupation number basis, is calculated fast [13] (After all, in Markov chain VMC only relative probabilities are required, which need not be normalized.)

---

[1]Hamiltonians with two-body off-diagonal terms such as indirect Coulomb interaction would require a lowrank update for all "two-hop states" that arise from a common reference state.

In the autoregressive ansatz, on the other hand, the issue is that changing the position of a single particle changes the conditional probabilities for all subsequent particles in the given ordering (see Fig. 8). As far as the Jastrow factor is concerned this is not a problem as the probability $p_{\text{Jastrow}}(\mathbf{n}^{(\beta)})$ is obtained by a *single* pass through the MADE network [21], which is much cheaper than the sampling step, which requires $N_p$ passes with $N_p$ the number of components (i.e. particles). The Slater sampler encoding $p_{\text{SD}}(\mathbf{n}^{(\beta)})$, however, has an internal state, and probability density estimation of an arbitrary occupation number state is as costly as sampling a state since all components have to be processed in order to update the internal state (the matrix $\tilde{X}^{-1}[k]$ in Eq. (29)) iteratively. Then, the evaluation of the kinetic energy would become the bottleneck of the algorithm: Since state $|\alpha\rangle$ is connected by the kinetic energy operator to $\mathcal{O}(N_p)$ states $|\beta\rangle$, evaluating the kinetic energy would be $\mathcal{O}(N_p)$ times more costly than the sampling of the reference state $|\alpha\rangle$, if $\langle\beta|\psi_\theta\rangle = \text{sign}(\beta)\sqrt{|\langle\beta|\psi_\theta\rangle|^2}$ needed to be re-evaluated for each "one-hop state" $|\beta\rangle$. Furthermore, as already mentioned, *all* conditional probabilities need to be calculated (not just at the sampled positions) since we need to be able to obtain the normalization constant $\mathcal{N}(\mathbf{n}^{(\beta)})$, see Eq. (35).

### 2.5.2 Lowrank updates from reference state

The goal is to compute the conditional probabilities in "one-hop"-state $|\beta\rangle$ based on the conditional probabilities of the reference state $|\alpha\rangle$ without recomputing any determinants. Assume that in state $|\alpha\rangle$ the $(k-1)$-th particle is located at $i_{k-1} \equiv m$ and the the $k$-th particle at $i_k \equiv m+l$. In between there are by definition no particles. With $k$ the component (i.e. $k$-th particle) being sampled and $i_k$ the position in question the conditional probability reads

$$p_{\text{cond}}^{(\alpha)}(k, i_k = m+l) \equiv p(n_{m+l}=1|n_1^{(\alpha)}, n_2^{(\alpha)}, \ldots, n_m^{(\alpha)}, n_{m+1}=0, \ldots, n_{m+l-1}=0, n_{m+l}=1)$$

$$= (-1) \frac{\det\left(G_{M+L,M+L} - N_{M+L}^{(\alpha)}\right)}{\det\left(G_{M,M} - N_M^{(\alpha)}\right)}, \tag{42}$$

where $M = \{1, 2, \ldots, m\}$ and $L = \{m+1, \ldots, m+l\}$ indicate ordered sets of site indices, $G_{M,M}$ is the corresponding submatrix of the equal-time Green's function and $N_M^{(\alpha)} = \text{diag}(n_1^{(\alpha)}, n_2^{(\alpha)}, \ldots, n_m^{(\alpha)})$ is a diagonal matrix whose entries are the occupation numbers up to site $m$ in basis state $|\alpha\rangle$. In $N_{M+L}^{(\alpha)} = N_M^{(\alpha)} \oplus \text{diag}(0, 0, \ldots, 0, +1)$ the occupation numbers on sites $m+1$ to $m+l-1$ are all set to zero and only site $m+l$ is occupied.

It is convenient to introduce for the conditional probability and for the ratio of numerator and denominator determinants in Eq. (42) the short-hand notation

$$p_{\text{cond}}^{(\alpha)}(k, i_k) = (-1) \frac{\det(G_{\text{num}}^{(\alpha)})}{\det(G_{\text{denom}}^{(\alpha)})}. \tag{43}$$

For later use, let us also highlight the block matrix structure of the expression Eq. (42):

$$\frac{\det\begin{pmatrix} A & B \\ C & D \end{pmatrix}}{\det(A)}, \tag{44}$$

where

$$A = [G - N]_{M,M}, \tag{45a}$$

$$B = [G]_{M,L}, \tag{45b}$$

$$C = B^T, \tag{45c}$$

$$D = [G - N]_{L,L}, \tag{45d}$$

$|\alpha\rangle \quad p_{cond}^{(\alpha)}(k=4, i_k=7)$

$= p(n_7 = 1 | n_1^{(\alpha)} = 1, n_2^{(\alpha)} = 0, n_3^{(\alpha)} = 1, n_4^{(\alpha)} = 1, n_5^{(\alpha)} = 0, n_6 = 0)$

$$= \frac{\det(\blacksquare\!\blacksquare\!\blacksquare\!+\!\!+\!\bullet\!+\!\boxtimes)}{\det(\blacksquare\!\blacksquare\!\blacksquare)}$$

$|\beta\rangle \quad p_{cond}^{(\beta)}(k=4, i_k=7)$

$= p(n_7 = 1 | n_1^{(\beta)} = 1, n_2^{(\beta)} = 1, n_3^{(\beta)} = 0, n_4^{(\beta)} = 1, n_5^{(\beta)} = 0, n_6 = 0)$

$$= \frac{\det(\blacksquare\!\blacksquare\!\blacksquare\!+\!\!+\!\bullet\!+\!\boxtimes)}{\det(\blacksquare\!\blacksquare\!\blacksquare)}$$

Figure 9: Graphical representation of the ratio of determinants in Eq. (42) and Eq. (46) for a reference state $|\alpha\rangle$ and a related one-hop state $|\beta\rangle$. In this example the conditional probabilities are for placing the fourth ($k=4$) out of five particles on nine sites. Given e.g. $p^{(\alpha)}(k=4, i_k=7)$, one can obtain $p^{(\beta)}(k=4, i_k=7)$ using a lowrank update of the numerator and denominator determinant in which the diagonal elements $(r, r)$ and $(s, s)$ are changed (see main text).

and $M$ and $L$ are the index sets defined above.

For describing the lowrank update systematically a graphical representation of conditional probabilities is useful. As already said, we assume that the conditional probability $p_{\text{cond}}^{(\alpha)}(k, i_k)$ for the $k$-th component in the reference state $\mathbf{n}^{(\alpha)}$ has been calculated at position $i_k = m + l$. In another occupation number state $\mathbf{n}^{(\beta)}$ the expression of the conditional probability for occupation of the $(m + l)$-th site differs only in the diagonal matrices $N_M^{(\beta)}$ and $N_{M+L}^{(\beta)} = N_M^{(\beta)} \oplus \text{diag}(0, 0, \ldots, 0, 1)$ while the submatrices of the equal-time Green's function are the same:

$$p_{\text{cond}}^{(\beta)}(k, i_k = m + l) = (-1) \frac{\det\left(G_{M+L,M+L} - N_{M+L}^{(\beta)}\right)}{\det\left(G_{M,M} - N_M^{(\beta)}\right)} . \tag{46}$$

This motivates the graphical representation of the conditional probabilities in terms of the entries of the diagonal matrices $N_{M+L}^{(\gamma)}$ and $N_M^{(\gamma)}$, $\gamma = \{\alpha, \beta\}$, in the numerator and denominator determinant, respectively. In Fig. 9, lattice sites, which are counted from left to right in the chosen fermion ordering, are denoted as boxes, and black points indicate occupied positions. A green point indicates the position for which the conditional probability is to be calculated. The green line indicates the support $i_k \in \{i_{\min}(k), i_{\max}(k)\}$ of the conditional probabilities of component $k$. In the example in Fig. 9 we consider the conditional probabilities for particle number $k = 4$ out of five particles on nine sites. In accordance with the ordering constraint Eq. (11), the last position $i = 9$ marked by a cross is excluded from the support since a fifth particle still needs to fit in somewhere so that this position cannot be occupied by the fourth particle. In the example in Fig. 9, the state $|\beta\rangle$ is obtained from $|\alpha\rangle$ by letting a particle hop from position $r = 4$ to position $s = 2$, i.e. $n_r^{(\alpha)} - 1 = n_r^{(\beta)}$ and $n_s^{(\alpha)} + 1 = n_s^{(\beta)}$. Therefore, the numerator and denominator matrices differ only in the diagonal entries $(r, r)$ and $(s, s)$ as follows:

$$\left(G_{\text{num}}^{(\beta)}\right)_{r,r} = \left(G_{\text{num}}^{(\alpha)}\right)_{r,r} + 1, \tag{47}$$

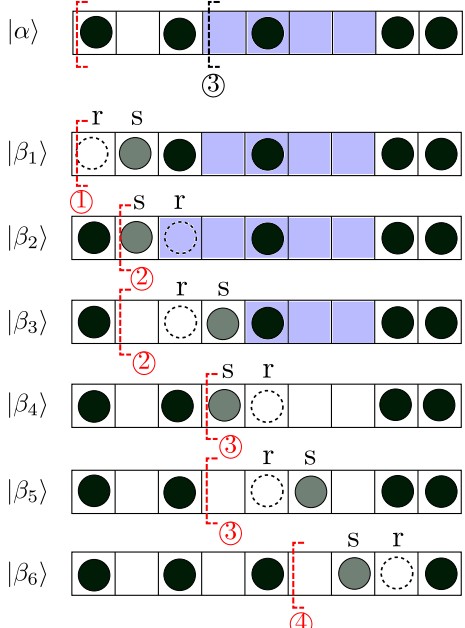

Figure 10: **Ordering of one-hop states.** A reference state $|\alpha\rangle$ and the associated one-hop states $|\beta_j\rangle \sim \hat{c}^\dagger_{s(j)}\hat{c}_{r(j)}|\alpha\rangle$, $j = 1,\ldots,6$, generated by the kinetic operator for a chain with nearest-neighbour hopping. Red dashed brackets with circled numbers $\text{(k)} = k_{\text{copy}[\beta]}$ indicate that conditional probabilities up to the $k$-th component (inclusive) can be copied from the reference state. Lattice positions shaded in blue indicate the support for the calculation of conditional probabilities for the example of the component $k = 3$. For the states $|\beta_4\rangle, |\beta_5\rangle, |\beta_6\rangle$ the conditional probabilities for the position of the third particle can be copied from state $|\alpha\rangle$. Note that for $|\beta_2\rangle$ the support of $p^{(\beta_2)}_{\text{cond}}(k = 3, i_k)$ is larger than that of $p^{(\alpha)}_{\text{cond}}(k = 3, i_k)$.

and similarly

$$\left(G^{(\beta)}_{\text{num}}\right)_{s,s} = \left(G^{(\alpha)}_{\text{num}}\right)_{s,s} - 1\,. \tag{48}$$

Eq. (47) describes the removal of a particle at position $r$ and Eq. (48) the addition of a particle at position $s$. Since $G^{(\beta)}_{\text{num}}$ and $G^{(\beta)}_{\text{denom}}$ differ from $G^{(\alpha)}_{\text{num}}$ and $G^{(\alpha)}_{\text{denom}}$ only in two diagonal matrix elements, their determinants can be updated from those of the $\alpha$-states using a low-rank update with $\mathcal{O}(1)$ operations. This is discussed in the next section.

### 2.5.3 Correction factor

The "onehop states" $|\beta\rangle$ are ordered according to the smallest particle position in which they differ from the reference state. The rationale is that, if a onehop state agrees with the reference state up to the $k$-th particle position, the conditional probabilities of the reference state up to the $k$-th particle can simply be copied. The largest $k$ up to which (inclusive) conditional probabilities for state $|\beta\rangle$ can be copied from state $|\alpha\rangle$ is called $k_{\text{copy}}[\beta]$. Fig. 10 illustrates the ordering of one-hop states.

Let $i_k[\beta]$ denote the position of the $k$-th particle in the one-hop state $|\beta\rangle$, which is assumed to arise from the reference state $|\alpha\rangle$ by a particle hopping from position $r \equiv r[\beta]$ to position $s \equiv s[\beta]$. We wish to compute the conditional probability of the $k$-th particle in the one-hop state $|\beta\rangle$ based on the conditional probabilities for the $k$-th particle in the reference state $|\alpha\rangle$. In the simplest case where $r, s < i_k$ (e.g. $|\beta_1\rangle$ in Fig. 10), both the numerator and denominator matrix in Eq. (46) need to be corrected in the positions $r$ and $s$. This can be achieved by

a lowrank update of the corresonding inverse matrices. Let $G^{(\alpha)}$ denote either $G^{(\alpha)}_{\text{num}}[k,i]$ or $G^{(\alpha)}_{\text{denom}}[k]$ at a given sampling step (sampling whether the $k$-th particle is to be placed at position $i$) of the reference state $|\alpha\rangle$ and $G^{(\beta)}$ the corresponding matrices in the reference state $|\beta\rangle$. The lowrank update amounts to

$$G^{(\beta)}_{r,r} = G^{(\alpha)}_{r,r} + 1 \quad \text{"remove particle" at } r\,,$$
$$G^{(\beta)}_{s,s} = G^{(\alpha)}_{s,s} - 1 \quad \text{"add particle" at } s\,,$$

which is realized by

$$G^{(\beta)} = G^{(\alpha)} + U^{(r,s)}\left(V^{(r,s)}\right)^T\,, \tag{49}$$

with $m \times 2$ ($m$-th position being sampled) matrices $U^{(r,s)} = (\hat{e}_r|\hat{e}_s)$ and $V^{(r,s)} = (\hat{e}_r|-\hat{e}_s)$ containing unit vectors as columns. By the generalized determinant lemma

$$\det(G^{(\beta)}) = \det(G^{(\alpha)} + U^{(r,s)}(V^{(r,s)})^T) \tag{50}$$

$$= \det\left(\underbrace{\mathbb{1}_2 + (V^{(r,s)})^T G^{(\alpha)-1} U^{(r,s)}}_{C}\right) \times \det G^{(\alpha)}\,. \tag{51}$$

The "capacitance matrix" C is

$$C = \begin{pmatrix} 1 + (G^{(\alpha)-1})_{r,r} & (G^{(\alpha)-1})_{r,s} \\ -(G^{(\alpha)-1})_{s,r} & 1 - (G^{(\alpha)-1})_{s,s} \end{pmatrix}\,, \tag{52}$$

and its determinant gives the correction factor to the determinants in the update $G^{(\alpha)} \to G^{(\beta)}$:

$$\kappa^{(r,s)} \equiv \frac{\det(G^{(\beta)})}{\det(G^{(\alpha)})} = \det(C)$$
$$= (1 + (G^{(\alpha)-1})_{r,r})(1 - (G^{(\alpha)-1})_{s,s}) + (G^{(\alpha)-1})_{r,s}(G^{(\alpha)-1})_{s,r}\,. \tag{53}$$

Applying the lowrank update to both the numerator and denominator matrix the correction factor connecting the conditional probabilities in the reference state $|\alpha\rangle$ to the conditional probabilities of state $|\beta\rangle$ is obtained as

$$p^{(\beta)}_{\text{cond}}(k,i) = \frac{\kappa^{(r,s)}_{\text{num}}}{\kappa^{(r,s)}_{\text{denom}}} \times p^{(\alpha)}_{\text{cond}}(k,i)$$

$$= \frac{(1 + (G^{(\alpha)-1}_{\text{num}})_{r,r})(1 - (G^{(\alpha)-1}_{\text{num}})_{s,s}) + (G^{(\alpha)-1}_{\text{num}})_{r,s}(G^{(\alpha)-1}_{\text{num}})_{s,r}}{(1 + (G^{(\alpha)-1}_{\text{denom}})_{r,r})(1 - (G^{(\alpha)-1}_{\text{denom}})_{s,s}) + (G^{(\alpha)-1}_{\text{denom}})_{r,s}(G^{(\alpha)-1}_{\text{denom}})_{s,r}} \times p^{(\alpha)}_{\text{cond}}(k,i)\,. \tag{54}$$

For this to work, the inverses of the numerator and denominator matrices of the reference state $|\alpha\rangle$, $G^{(\alpha)-1}_{\text{num}}$ and $G^{(\alpha)-1}_{\text{denom}}$, need to be kept and updated iteratively during the component-wise sampling. As a result of the lowrank updates the conditional probabilities $p^{(\beta)}_{\text{cond}}(k,i)$ for all states $|\beta\rangle$ connected to $|\alpha\rangle$ by the hopping of a single particle can be calculated simultaneously with $p^{(\alpha)}_{\text{cond}}(k,i)$. With the lowrank update, the relative overhead of calculating the local kinetic energy $E^{(\text{kin, loc})}_{\theta}(\alpha)$ compared to the sampling or probabilitiy estimation of $|\alpha\rangle$ is only a constant factor which approaches $\mathcal{O}(1)$ asymptotically for large system sizes (see Fig. 11). Without it, the calculation of the local energy would be $\mathcal{O}(N_p)$-times slower. Therefore this relatively complicated low-rank update is essential to the efficiency of the proposed autoregressive Slater-Jastrow ansatz. Only the simplest case "remove-r-add-s" has been discussed here, all other details can be found in Appendix E.

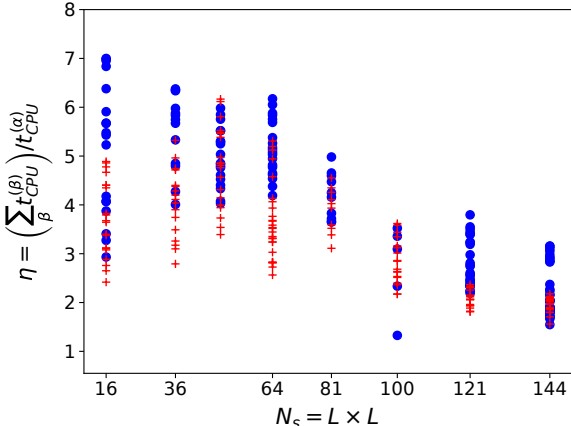

Figure 11: Relative overhead $\eta = \left(\sum_\beta t_{\text{CPU}}^{(\beta)}\right)/t_{\text{CPU}}^{(\alpha)}$ of CPU time for computing all conditional probabilities needed to obtain $p_{\text{SD}}(\mathbf{n}^{(\beta)})$ for states $|\beta\rangle$ connected to the reference state $|\alpha\rangle$ by the kinetic operator via lowrank updates compared to the CPU time needed for calculating the conditional probabilities and $p_{\text{SD}}(\mathbf{n}^{(\alpha)})$ for the reference state $|\alpha\rangle$ itself. As the cost of calculating the conditional probabilities $p_{\text{cond}}^{(\alpha)}$ for the reference state $|\alpha\rangle$ grows with system size $N_s = L^2$, the cost of the lowrank update for all states $|\beta\rangle$ is increasingly amortized, although the number of states $|\beta\rangle$ scales as $\sim N_p$ (The scatter indicates different randomly chosen reference states $|\alpha\rangle$; blue dots: half-filling with particle number $N_p = \lfloor L^2/2 \rfloor$; red crosses: quarter filling with $N_p = \lfloor L^2/4 \rfloor$; $V/t = 6$).

## 2.6 Optimization and simulation details

The optimization of the variational parameters is done with the stochastic reconfiguration (SR) method [69–71], which is also known as natural gradient descent [72, 73]. In the update of the variational parameters at optimization step $t$

$$\theta_p^{(t+1)} = \theta_p^{(t)} - \eta S^{-1} \frac{\partial \langle E_\theta \rangle}{\partial \theta_p}, \tag{55}$$

the stochastic gradient

$$g_p \equiv \frac{\partial \langle E_\theta \rangle}{\partial \theta_p} = \langle\langle O_p \mathcal{H} \rangle\rangle \tag{56}$$

is preconditioned by the inverse of the Fisher information matrix $S_{p,p'} = \langle\langle O_p O_{p'} \rangle\rangle$. In these expressions, the logarithmic derivative operator is defined as $O_p(x) = \frac{\partial}{\partial \theta_p} \log \psi_\theta(x)$ and the connected correlators $\langle\langle AB \rangle\rangle = \langle AB \rangle - \langle A \rangle \langle B \rangle$ are estimated stochastically by averaging over a batch of samples. Stochastic reconfiguration is very effective in thinning out redundant parameters and dealing with possible vastly different curvatures of the loss landscape that arises from the cooptimization of the Jastrow factor and the orbitals of the Slater determinant (see Appendix C). The autoregressive Jastrow factor is not translationally invariant and no other symmetries are imposed so that the number of variational parameters increases very quickly with system size like $N_{\text{param}} \sim \frac{1}{2}(N_p N_s)^2$. In order to circumvent the construction and storage of the matrix $S$, which is quadratic in $N_{\text{param}}$, the linear system $\eta \mathbf{g} = -S\left(\theta^{(t+1)} - \theta^{(t)}\right)$ is solved using the conjugate gradient method with lazy evaluation of matrix-vector products [74].

Redundant parametrization of the variational ansatz causes the covariance matrix $S$ to be non-invertible. In our case the Jastrow factor is heavily over-parametrized. Therefore a regularization of $S$ is required, which is accomplished by rescaling [71] the diagonal matrix elements and with a diagonal shift [2]. For SR the learning rate was $\eta = 0.2$.

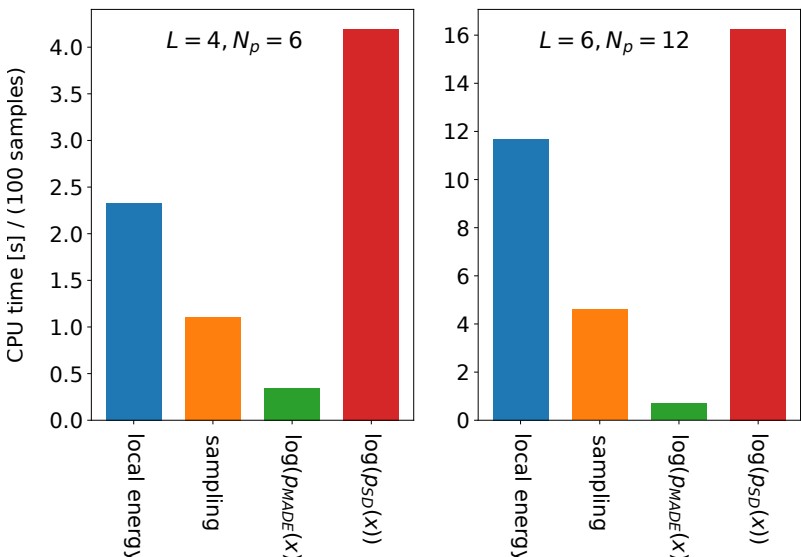

Figure 12: CPU time per 100 samples (in seconds) of the most important computational tasks with cooptimization of the Slater determinant.

Alternatively to SR we use the reinforcement loss

$$\nabla_\theta \mathcal{L} = \sum_{x \sim |\psi_\theta(x)|^2}^{M} \nabla_\theta \log(\psi_\theta(x)) \left[ E_\theta^{(\text{loc})}(x) - E_{\theta,\text{avg}} \right], \qquad (57)$$

where the gradient acts only on the wavefunction. Using stochastic gradient descent, the gradients are estimated over a batch of $M \approx 200-300$ samples and $E_{\theta,\text{avg}} = \frac{1}{M} \sum_{x \in \text{batch}}^{M} E_\theta^{(\text{loc})}(x)$. Backpropagation on this loss function $\mathcal{L}$ directly reproduces the gradients of the average energy Eq. (56) and thus allows to employ fine-tuned optimizers such as Adam [75] and the learning rate schedulers available in the PyTorch [64] machine learning framework.

An infinite variance of the local energy [76] is a known issue in fermionic QMC. It is related to the presence of nodes where the wavefunction can change sign during the optimization; a small value of $|\psi_\theta(x)|$ in Eq. (6) can make the calculation of $E_\theta^{(loc)}(x)$ unstable. Following Ref. [37], the local energy is clipped when calculating gradients in Eq. (57), but not when calculating the average local energy.

Fig. 12 shows that the largest part of the computation time is due to the inference of samples on the Slater determinant, $\log(p_{\text{SD}}(x))$, which is caused by the iterative procedure of calculating conditional probabilities (see Sec. 2.3) and the need to calculate all conditional probabilities (rather than just at the actually sampled positions) for the purpose of normalization. The second-largest contribution comes from the calculation of the local energy. In comparison, the running time for inference of the bosonic probabilities given by the MADE neural network is negligible.

The diagram is only meant to give an overall indication of the relevance of optimizing certain computational tasks; the CPU timings for different tasks are not disjoint.

## 3 Benchmark results

The implementation of the autoregressive Slater-Jastrow VMC ansatz (arSJVMC) is benchmarked on the $t-V$ model of $N_p$ spin-polarized fermions on a square lattice of $N_s = L^2$ sites

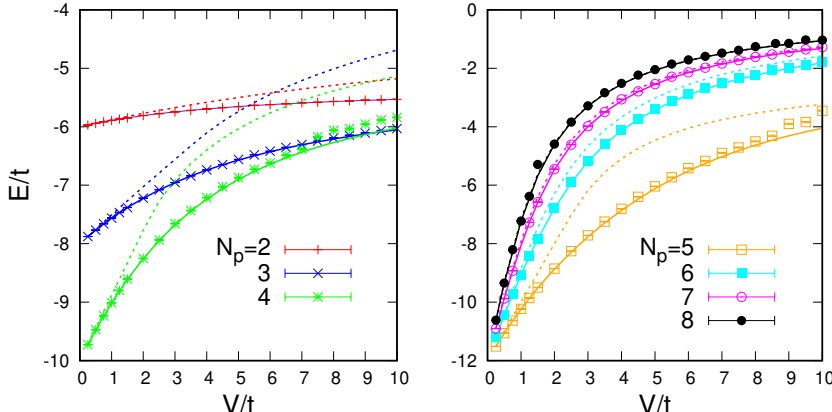

Figure 13: Comparison of the ground state energy obtained from arSJVMC (symbols) with exact diagonalization (continuous lines) and the Hartree-Fock solution (dashed lines) on a $4 \times 4$ system with particle number varying from $N_p = 2$ to $N_p = 8$.

with periodic boundary conditions:

$$H_{tV} = -t \sum_{\langle i,j \rangle} \left( c_i^\dagger c_j + c_j^\dagger c_i \right) + V \sum_{\langle i,j \rangle} n_i n_j \,, \tag{58}$$

where $\langle i,j \rangle$ denotes nearest neighbours and $t, V > 0$. Only at half filling and on a bipartite graph, the $t-V$ model with $V > 0$ does not have a sign problem in QMC [77–79] and unbiased simulations on large systems are possible. Away from half filling the phase diagram has been explored using various variational methods [7].

In order to assess the amount of correlation energy captured by the autoregressive Slater-Jastrow VMC, Fig. 13 compares the VMC results with the exact ground state energy and the energy from Hartree-Fock (HF) approximation of the Hamiltonian (58) (see e.g. Ref. [7]) on a $4 \times 4$ system. Around 98% of the correlation energy, defined as $E_{\text{HF}} - E_{\text{exact}}$, are recovered with a single (cooptimized) Slater determinant, except for large $V/t$ around quarter filling ($N_p = 4, 5$), where - apparently - static correlations become important. Interestingly, at half filling ($N_p = 8$), the Hartree-Fock solution provides already an extremely good approximation to the ground state such that the correlation energy is very small over the entire range of $V/t$-values.

In order to further verify the quality of the approximated ground state wavefunction we compare in Fig. 14 the density-density correlation function

$$C(\mathbf{r}) = \frac{1}{N_s} \sum_{\mathbf{i}} (\langle n_{\mathbf{i}} n_{\mathbf{i+r}} \rangle - \langle n_{\mathbf{i}} \rangle \langle n_{\mathbf{i+r}} \rangle) \,, \tag{59}$$

with results from exact diagonalization (ED). Following Ref. [7], the density-density correlations are also plotted as a function of graph distance

$$\tilde{C}(r) = \frac{1}{N_r} \sum_{\text{dist}(\mathbf{r})=r} C(\mathbf{r}) \,, \tag{60}$$

where $r = \text{dist}(\mathbf{r}) = |r_x| + |r_y|$ is the Manhattan distance and $N_r$ is the number of points with given distance $r$.

The Lanczos exact diagonalization was carried out using the Quspin package [80, 81], restricting to the momentum sectors which contain the ground state. Correlation functions were averaged over degenerate ground states in momentum sectors related by point group

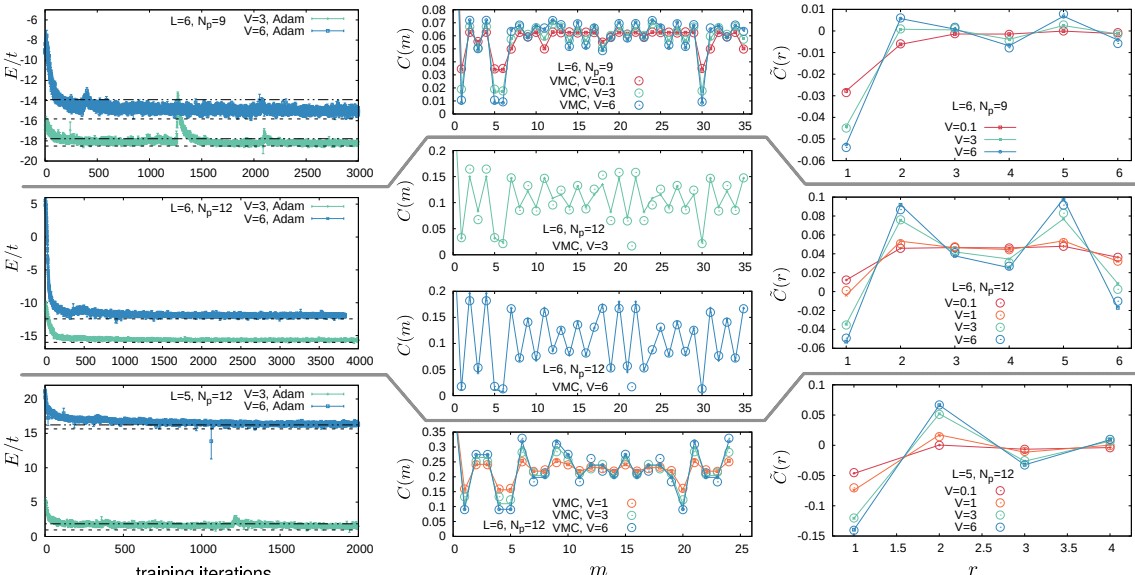

Figure 14: Left column: Convergence of the energy (with Adam used as an optimizer, batchsize is 200 samples). Dashed and dashed-dotted lines denote the exact energies of the ground and first excited state, respectively. Middle column: Density-density correlation function, where the variable $\mathbf{r} = (r_x, r_y)$ has been flattened such that $C(m) = C(\mathbf{r})$ with $m = r_x + r_y L$. Continuous lines are exact results from ED. Right column: Density-density correlations as a function of graph distance.

symmetry. As it involves only a single Hartree-Fock Slater determinant, for open-shell systems, the VMC ansatz does not necessarily have well-defined quantum numbers.

The orbitals of the Slater determinant are cooptimized so as to find the best single-determinant wavefunction in the presence of the Jastrow factor. The evolution of the optimized Slater determinant relative to the original Hartree-Fock Slater determinant is quantified through a measure of the change of the sign structure [4]

$$\frac{\sum_x |\psi_{\mathrm{HF}}(x)|^2 \mathrm{sign}(\psi_{\mathrm{HF}}(x))\mathrm{sign}(\psi_\theta(x))}{\sum_x |\psi_{\mathrm{HF}}(x)|^2}, \tag{61}$$

and the overlap of the initial and the optimized wavefunctions

$$\langle \psi_{\mathrm{HF}} | \psi_{\mathrm{SD},\theta} \rangle. \tag{62}$$

The evolution of these quantities during optimization is shown in Fig. 16(b) for a larger system ($L = 6$). In order to verify the effectiveness of the cooptimization, the Slater determinant has been initialized with a HF solution which is not fully converged so that part of the HF orbital optimization is transferred to the VMC loop. As Figs. 15 and 16 show, with cooptimization, the ground state is recovered even with a poor starting point for the mean-field wavefunction. For small systems ($L = 4$), only the overlap $\langle \psi_{\mathrm{HF}} | \psi_\theta \rangle$ changes during optimization (inset Fig. 15) whereas the measure of the sign structure stays pinned to 1. For larger system (Fig. 16), the sign structure also changes considerably after an initial plateau around 1. On the other hand, with a fully self-consistent HF Slater determinant as a starting point, the beneficial effect of cooptimizing the orbitals is much less significant. It should be pointed out that the computational cost of automatic differentiation for optimizing the orbitals of the Slater determinant (i.e. calculation of gradients of $\log(\psi_\theta(x))$ in Eq. (57)) is approximately an order of magnitude larger than the cost of automatic differentiation for calculating gradients with respect to

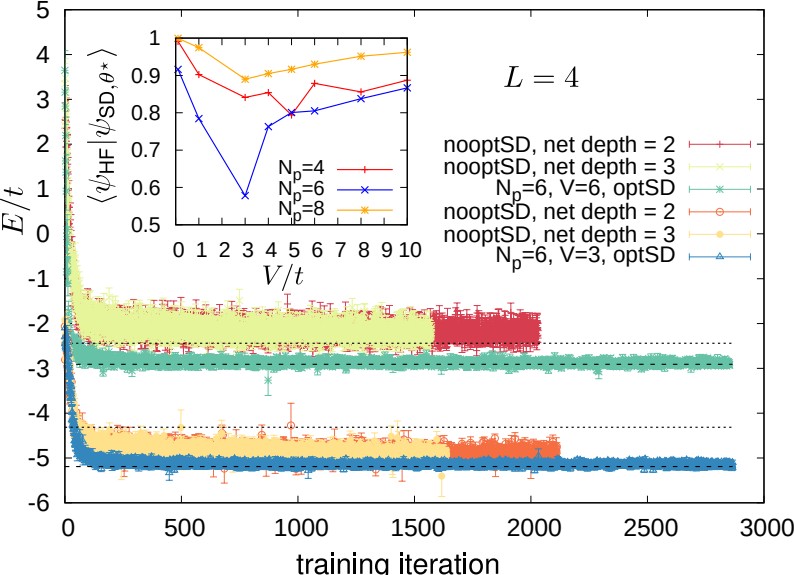

Figure 15: Cooptimization of the orbitals of the Slater determinant together with the Jastrow factor allows for convergence to the ground state even if the initial HF Slater determinant is poorly converged. optSD (nooptSD) denotes that the Slater determinant was (not) cooptimized. Dashed (dotted) lines indicate the exact ground state energy (energy of the first excited state). Adam was used as an optimizer.

parameters of the MADE neural network alone. This is due to the iterative process by which the conditional probabilities under the Slater determinant are calculated.

The largest Hilbert space dimension for the test systems is $\binom{36}{15} \approx 5.6 \times 10^9$ (see Fig. 16). Due to memory constraints, for this system size no exact ground state energy was available to us, and we use the correlation function from Ref. [7] as a benchmark (see inset in Fig. 16), finding excellent agreement. Best results over five random seeds and variance extrapolation of the energies (Fig. 17) for this set of simulations are shown in Tab. 1 for a range of interactions and filling fractions. The relative error $\Delta E = |E_{\text{arSVMC}} - E_{\text{exact}}|/|E_{\text{exact}}|$ is always below $1-2\%$, which is consistent with an ansatz limited by the sign structure of a single Slater determinant. Such an error is comparable to other works [2, 51].

Tabs. 2,3,4 compare energies obtained with our arSJVMC to energies from a Slater-Jastrow ansatz represented by a restricted Boltzmann machine (RBM) [2] and other methods from the comprehensive variational benchmark published in Ref. [82, 84]. The conclusion from this comparison is that, in terms of accuracy, the arSJVMC is on par with a simple RBM-Slater-Jastrow ansatz with a single Slater determinant, while a more advanced RBM-Slater-Jastrow ansatz with symmetry projection and backflow correlations can reach almost exact ground state energies. For the half-filled $t-V$ model on a bipartite lattice, comparison can also be made with the unbiased continuous time quantum Monte Carlo method [83], which is free of the sign problem, see Tab. 4. It turns out that for $L = 8, N_p = 32$ at $V/t = 4$, the arSJVMC ansatz is not lower than the HF energy, which is due to difficulties in the training rather than the expressibility of the ansatz. Notably, the CTQMC energy is also higher than the HF energy. As an explanation, the HF energies in Tab. 4 and Fig. 13 suggest that for the $t-V$ model on the square lattice, the HF approximation is already extremely accurate at half filling and therefore difficult to surpass. As the Jastrow factor is initialized randomly (i.e. it is not an identity), the arSJVMC ansatz is not guaranteed to reach a lower energy than a single Slater determinant in the HF approximation.

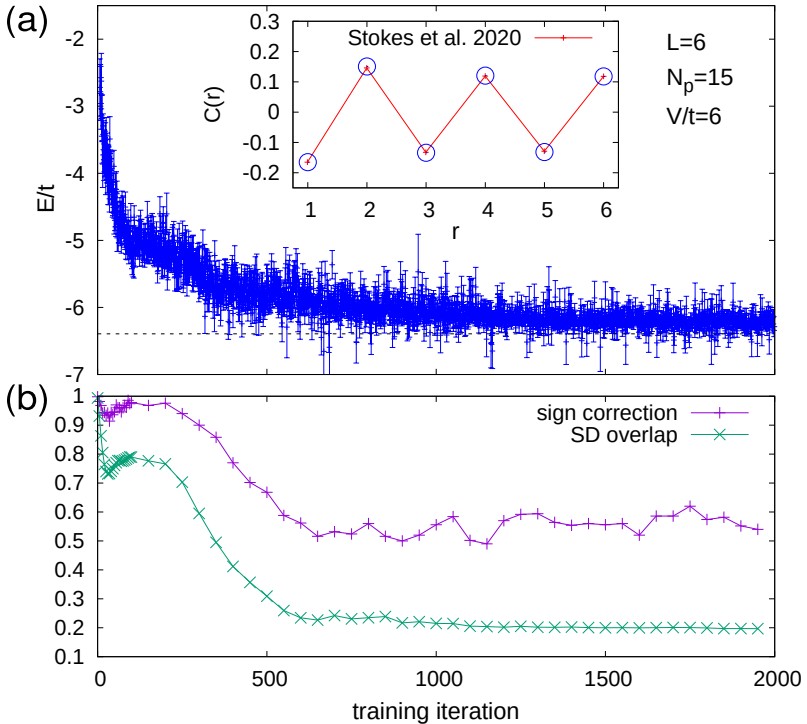

Figure 16: (a) Convergence of energy for system size $L = 6$ with $N_p = 15$ particles and $V/t = 6$, starting from a poorly converged (i.e. not self-consistent) HF Slater determinant. The dashed line is the energy from variance extrapolation (not the exact ground state energy). The density-density correlations as a function of graph distance agree with data from Ref. [7] (inset). The sign structure of the optimized Slater determinant differs considerably from the initial one: (b) shows the evolution during training of the Slater determinant overlap Eq. (62) and the measure for the change in sign structure, Eq. (61).

Table 1: Benchmark of energies for $L = 6$ and $N_p = 9, 12$ (second three rows) and $N_p = 15$ (last row, where the Hilbert space was too large for exact diagonalization). The second and third column give the exact ground state energy and the Hartree-Fock (HF) approximation. The fourth column shows the best variational energies accross five random seeds and the the fifth column the corresponding variance extrapolation. The relative error after variance extrapolation is on the order of (1-2)%.

| $(N_p, V/t)$ | exact | HF | arSJVMC (best) | arSJVMC (extrap.) |
|---|---|---|---|---|
| (9, 1.0) | -21.747707 | -21.499999 | -21.732(3) | – |
| (9, 3.0) | -18.510441 | -16.499999 | -18.370(8) | – |
| (9, 6.0) | -15.820226 | -12.195724 | -15.40(2) | -15.58(9) |
| (12, 1.0) | -22.171946 | -21.599772 | -22.09(1) | – |
| (12, 3.0) | -16.023298 | -14.340569 | -15.67(2) | -15.83(1) |
| (12, 6.0) | -12.45152 | -10.414462 | -12.13(3) | -12.19(4) |
| (15, 6.0) | – | -6.1515321 | -6.27(2) | -6.39(5) |

Table 2: Comparison of our autoregressive Slater-Jastrow ansatz (arSJVMC) with Slater-Jastrow (SJ) and Slater-Jastrow-Backflow (SJB) wavefunctions realized with a restricted Blotzmann machine (RBM) [82], with and without symmetry-projection onto the ground state of zero momentum ($K = 0$). Hartree-Fock (HF) and Lanczos exact diagonalization (ED) indicate the expected upper and lower bounds of the variational energy.

| Method | System ($L = 4, N_p = 5$) | | | |
| | $V/t = 0.01$ | $V/t = 0.1$ | $V/t = 1$ | $V/t = 10$ |
| --- | --- | --- | --- | --- |
| HF | −11.980000 | −11.800000 | −10.000000 | −3.232766 |
| arSJVMC (this work) | −11.97996(5) | −11.8019(6) | −10.238(1) | −3.93(1) |
| Lanczos ED (QuSpin) | −11.980026 | −11.802611 | −10.240650 | −4.052055 |
| SJ (RBM) | −11.9800(4) | −11.80259(4) | −10.2394(2) | −3.778(4) |
| SJ (RBM), $K = 0$ | | −11.802606(4) | −10.2399(1) | −3.79(2) |
| SJB (RBM), $K = 0$ | −11.980025(1) | −11.802611(4) | −10.24062(4) | −4.02(2) |

Table 3: System: $L = 6, N_p = 13$; see Tab. 2 for details.

| Method | System ($L = 6, N_p = 13$) | | | |
| | $V/t = 0.01$ | $V/t = 0.1$ | $V/t = 1$ | $V/t = 10$ |
| --- | --- | --- | --- | --- |
| HF | −27.933333 | −27.333333 | −21.333333 | −6.908615 |
| arSJVMC (this work) | −27.9332(4) | −27.332(1) | −21.85(1) | −7.23(3) |
| Lancos ED (QuSpin) | −27.93340531 | −27.34054415 | −22.07737235 | −7.92802624 |
| SJ (RBM), $K = 0$ | −27.933406(5) | −27.34055(5) | −22.017(3) | −6.16(8) |
| SJB (RBM), $K = 0$ | −27.93340(1) | −27.340544(3) | −22.0749(2) | −7.566(7) |

Table 4: Comparison with DMRG and continuous time quantum Monte Carlo (CTQMC) [82,83] at half filling, where the sign problem can be avoided. For $V/t = 2$ and 4, the VMC energy falls short of even the Hartree-Fock energy, which is, however, in the present case of half filling already extremely close to the exact ground state energy (see also Fig. 13).

| Method | System ($L = 8, N_p = 32$) | | |
| | $V/t = 1$ | $V/t = 2$ | $V/t = 4$ |
| --- | --- | --- | --- |
| HF | −29.193934 | −18.506328 | −10.221101 |
| arSJVMC (this work) | −28.78(2) | −17.56(4) | −9.28(4) |
| CTQMC | −29.48(2) | −18.7(1) | −9.8(2) |
| DMRG ($\chi_{\max} = 4096$) | −29.380064 | −18.633949 | −10.248601 |

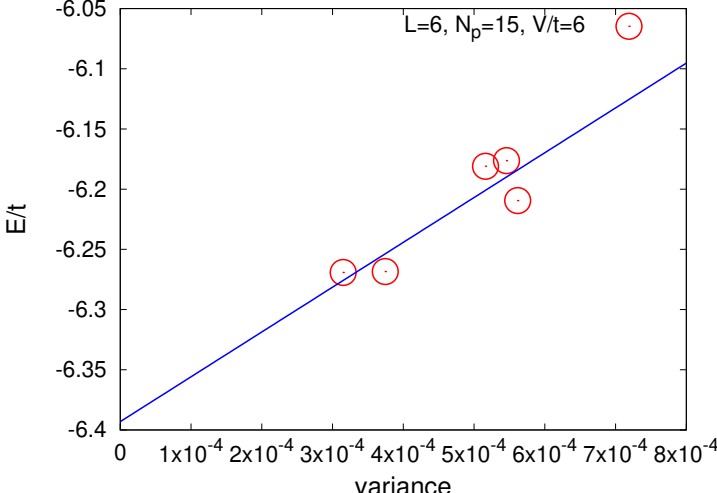

Figure 17: Variance extrapolation of the energy for the parameters of Fig. 16.

## 4 Outlook

A natural question is how corrections to the sign structure of the single Slater determinant can be incorporated into an autoregressive framework. Apart from using a separate neural network dedicated to sign corrections [7] (which does not affect the ability to directly sample from the ansatz [60]), there are well-established multireference ansätze. This includes the linear superposition of determinants that are built as particle-hole excitations from a common reference Slater determinant [67,85,86], Paffian pairing wave functions [13,87,88] and orbital backflow [4,89], where the orbitals of the Slater determinant depend on the configuration.

Multi-determinant wavefunctions with a small number of determinants (on the order of the system size) are useful as they allow for symmetry-projection [45,90]. The necessary low-rank updates [85] resemble those of Sec. 2.5. However, ultimately, for systematic improvement of the sign structure an exponentially large number of excited orthogonal Slater determinants needs to be included [91] for a sizable effect. A more economical ansatz is a Pfaffian pairing wavefunction or antisymmetrized geminal power (AGP) [49] which constitues a resummation of a certain subset of Slater determinants and provides a larger variational space at the computation cost of a single Slater determinant [13]. The normalized AGP wavefunction reads

$$|\psi_{\text{AGP}}\rangle = \frac{1}{\left(\frac{N_p}{2}\right)! \, 2^{N_p/2}} \left( \sum_{i,j=1}^{N_s} F_{ij} c_i^\dagger c_j^\dagger \right)^{N_p/2} |0\rangle \equiv |F\rangle \,, \tag{63}$$

where $F^T = -F$ is a pairing wavefunction. While the overlap of a Pfaffian with a single Slater determinant $|\alpha\rangle$, i.e. $\langle \alpha | F \rangle$, can be expressed in terms of a Pfaffian of $F$, which is all that is needed for performing sampling in Markov chain VMC [13], there is no known compact formula for the overlap of two different Pfaffian or AGP wavefunctions $\langle F | F' \rangle$. An AGP wavefunction can be written as a projection of a Hartree-Fock-Bogoliubov (HFB) wavefunction, which is a product of independent quasiparticles, onto a fixed particle-number sector, i.e. it is a linear combination of HFB states for which there is an efficient overlap formula [92,93] since Wick's theorem applies to each HFB state individually. However, Wick's theorem is not valid for the linear combination of HFB states and an overlap formula for different AGP states is not known to us. The absence of a computationally efficient expression for the marginal probabilities of Eq. (23) appears to be an obstacle to formulating an autoregresssive Pfaffian-Jastrow ansatz, which warrants further investigation.

Finally, incorporating a general backflow transformation into an autoregressive neural network naively would lead to a prohibitive computational cost scaling like $\mathcal{O}(N^5)$ rather than $\mathcal{O}(N^4)$ for neural network backflow in a Slater-Jastrow ansatz with Markov chain Monte Carlo sampling [4]. The reasoning is the following: With the backflow transformation affecting each entry of the Green's function in Eq. (23), low-rank updates are not possible and all determinants need to be calculated from scratch, which costs $\mathcal{O}(N^3)$. Calculating N conditional probabilities for one uncorrelated sample therefore costs $\mathcal{O}(N^4)$. The conditional probabilities need to be normalized because, although the probability distributions of the Slater sampler and the Jastrow factor are individually normalized, their product is not. This has implications for the inference step (when calculating local energy): Calculating the probability of some configuration is as expensive as sampling a configuration since we need all conditional probabilities for the purpose of normalization, not just those at the actually sampled positions. Therefore density estimation also costs $\mathcal{O}(N^4)$. When calculating local energy we need the probabilities of all states connected to the sampled state by the kinetic term. There are $\mathcal{O}(N)$ such states for nearest-neighbour hopping and density estimation for each one costs $\mathcal{O}(N^4)$. Therefore the overall cost for calculating the local energy is $\mathcal{O}(N^5)$.

With a view towards *ab initio* simulations, e.g. of small molecules, one needs to find an efficient way to evaluate the contribution of the (off-diagonal) Coulomb interaction to the local energy. This can be achieved by a low-rank update analogous to that for the local kinetic energy where the states $|\alpha\rangle$ and $|\beta\rangle$ can differ in up to four positions.

Another future direction aimed at improving the scalability [61] is the replacement of the MADE network by another autoregressive architecture such as the PixelCNN [24] or RNN [11,94] in order to reduce the number of variational parameters, which in the current approach scales like $N_{\text{param}} \sim N^4$ and may limit the achievable system sizes due to memory constraints.

## 5 Conclusion

In conclusion, we have presented an autoregressive Slater-Jastrow ansatz suitable for variational Monte Carlo simulation which allows for uncorrelated sampling while retaining the cubic scaling of the computational cost with system size. This comes at the price of implementing a complicated low-rank update for calculating the off-diagonal part of the local energy.

A number of challenges need to be addressed before the described method can be put to practical use: (i) Evaluating the probability of a configuration under the ansatz (density estimation) is as costly as sampling the configuration. Therefore symmetry projection, which consists in associating with a configuration some average of the probabilities of all symmetry related configurations [90], becomes forbiddingly expensive. (ii) There is an imbalance between the expressibility of the Jastrow factor and the Slater determinant part. The former is heavily over-parametrized, and the required memory to store the network parameters limits the simulatable system sizes to around $10 \times 10$. This can be remedied by choosing another autoregressive architecture such as PixelCNN or RNN. At the same time, calculating the conditional probabilities due to the Slater determinant and cooptimizing the orbitals takes up the bulk of the simulation time, which calls for further optimizations. While our method in its current implementation is slower than other neural network VMC schemes such as e.g. Slater-Jastrow-RBM, the formally cubic scaling holds the promise of ultimately accessing larger system sizes.

An implementation written in PyTorch is available in the code repository [95].

## Acknowledgment

SH thanks D. Luo for helpful discussions.

**Funding information**   The work of SH was supported by the Simons Collaboration on the Many Electron Problem. LW is supported by the Strategic Priority Research Program of the Chinese Academy of Sciences under Grant No. XDB30000000 and National Natural Science Foundation of China under Grant No. T2121001.

## A   Local one-body density matrix (OBDM)

For completeness this and the following section review a number of well-known relations for Slater determinants, see for instance [96]. A Slater determinant can be written as

$$|\psi\rangle = \prod_{n=1}^{N_p} \sum_{m=1}^{N_s} P_{m,n} c_n^\dagger |0\rangle, \tag{64}$$

with an $N_s \times N_p$ matrix $P$ whose columns contain the orthonormal single-particle eigenstates (*P-matrix representation*). Let $|\alpha\rangle$ and $|\psi\rangle$ denote two Slater determinants with the same number of particles and let $P_\alpha$ and $P_\psi$ be their P-matrices. Then the local Green's function is given as

$$G_{ij}^{(\alpha,\psi)} = \frac{\langle \alpha | c_i c_j^\dagger | \psi \rangle}{\langle \alpha | \psi \rangle} = \delta_{ij} - \left[ P_\psi \left( P_\alpha^T P_\psi \right)^{-1} P_\alpha^T \right]_{ij}. \tag{65}$$

The local one-body density matrix is

$$\mathcal{G}_{ji}^{(\alpha,\phi)} = \delta_{ij} - G_{ij}^{(\alpha,\phi)} = \left[ P_\psi \left( P_\alpha^T P_\psi \right)^{-1} P_\alpha^T \right]_{ij}. \tag{66}$$

**Proof:**

$$\langle \alpha | c_i c_j^\dagger | \psi \rangle = \det\left( \left( P_\alpha^{(i)'} \right)^T P_\psi^{(j)'} \right), \tag{67}$$

where the primed matrix $P_\alpha^{(i)'}$ arises from $P_\alpha$ by adding a particle at position $i$, i.e.

$$P_\alpha^{(i)'} = \left( P_\alpha \quad |\hat{e}_i \right), \quad P_\psi^{(j)'} = \left( P_\psi \quad |\hat{e}_j \right),$$

so that

$$\left( P_\alpha^{(i)'} \right)^T P_\psi^{(j)'} = \begin{pmatrix} P_\alpha^T P_\psi & \left( P_\alpha^T \right)_{:,j} \\ \left( P_\psi \right)_{i,:} & \delta_{ij} \end{pmatrix}. \tag{68}$$

Using Schur complementation of this block matrix its determinant is seen to be

$$\langle \alpha | c_i c_j^\dagger | \psi \rangle = \det\left( P_\alpha^T P_\psi \right) \cdot \left( \delta_{ij} - \sum_{k,l=1}^{N_p} \left( P_\psi \right)_{i,k} \left( \left( P_\alpha^T P_\psi \right)^{-1} \right)_{k,l} \left( P_\alpha^T \right)_{l,j} \right). \tag{69}$$

With $\langle \alpha | \psi \rangle = \det\left( P_\alpha^T P_\psi \right)$ the stated result Eq. (65) follows. $P_\alpha^T P_\psi$ is an $N_p \times N_p$ matrix, which needs to be inverted, and the number of operations for calculating all elements of the local Green's function is thus $\mathcal{O}(N_p^3) + \mathcal{O}(2N_p^2 N_s)$.

## B  Slater determinant overlap ratios

Let $P_\alpha$ and $P_\beta$ denote P-matrix representations of occupation number states related by a particle hopping from occupied position $r$ in $|\alpha\rangle$ to an unoccupied position $s$. $P_\beta$ is obtained from $P_\alpha$ by a lowrank update

$$P_\beta = (\mathbb{1}_{N_s} - \Delta(r,s))P_\alpha\Pi_{\text{sort}}, \tag{70}$$

with $\Delta(r,r) = \Delta(s,s) = 1$ and $\Delta(r,s) = \Delta(s,r) = -1$ and all other elements of $\Delta$ equal to zero. $\Pi_{\text{sort}}$ makes sure that the columns of $P_\beta$ are ordered according to increasing row index of particle positions. To illustrate this point, consider the following example with $[\alpha] = [0,1,0,1,1]$ and $[\beta] = [1,1,0,1,0]$, i.e. $|\beta\rangle$ arises from $|\alpha\rangle$ by a particle hopping from $r = 5$ to $s = 1$. The P-matrix representations of these Fock states and the factors connecting them are:

$$P_\alpha = \begin{pmatrix} 0 & 0 & 0 \\ 1 & 0 & 0 \\ 0 & 0 & 0 \\ 0 & 1 & 0 \\ 0 & 0 & 1 \end{pmatrix}, \quad (\mathbb{1}_5 - \Delta(r,s))P_\alpha = \begin{pmatrix} 0 & 0 & 1 \\ 1 & 0 & 0 \\ 0 & 0 & 0 \\ 0 & 1 & 0 \\ 0 & 0 & 0 \end{pmatrix}, \quad \Pi_{\text{sort}} = \begin{pmatrix} 0 & 1 & 0 \\ 0 & 0 & 1 \\ 1 & 0 & 0 \end{pmatrix}, \quad P_\beta = \begin{pmatrix} 1 & 0 & 0 \\ 0 & 1 & 0 \\ 0 & 0 & 0 \\ 0 & 0 & 1 \\ 0 & 0 & 0 \end{pmatrix}.$$

The ratio of overlaps of $|\alpha\rangle$ and $|\beta\rangle$ with an arbitrary Slater determinant $|\psi\rangle$ is then:

$$\begin{aligned} R = \frac{\langle\beta|\psi\rangle}{\langle\alpha|\psi\rangle} &= \frac{\det(P_\alpha^T(\mathbb{1}_{N_s} - \Delta(r,s))P_\psi)}{\det(P_\alpha^T P_\psi)} \times \det(\Pi_{\text{sort}}) \\ &= \det(\mathbb{1}_{N_p} - (P_\alpha^T P_\psi)^{-1}P_\alpha^T\Delta(r,s)P_\psi) \times \det(\Pi_{\text{sort}}) \\ &= \det(\mathbb{1}_{N_s} - \Delta(r,s)P_\psi(P_\alpha^T P_\psi)^{-1}P_\alpha^T) \times \det(\Pi_{\text{sort}}), \end{aligned} \tag{71}$$

where in the last step the identity $\det(\mathbb{1}_M + AB) = \det(\mathbb{1}_N + BA)$ for rectangular $M \times N$ and $N \times M$ matrices $A$ and $B$ has been used. From Eq. (66) one may recognize the local OBDM between Slater determinants $|\alpha\rangle$ and $|\phi\rangle$ so that:

$$R = \det(\mathbb{1}_{N_s} - \Delta(r,s)\left(\mathcal{G}^{(\alpha,\psi)}\right)^T) \times \det(\Pi_{\text{sort}}). \tag{72}$$

The transpose may be dropped since $\mathcal{G}^{(\alpha,\psi)}$ is hermitian. As $\Delta(r,s)$ has only four non-zero entries, the final result is

$$\frac{\langle\beta|\psi\rangle}{\langle\alpha|\psi\rangle} = (1 - \mathcal{G}_{r,r}^{(\alpha,\psi)} - \mathcal{G}_{s,s}^{(\alpha,\psi)} + \mathcal{G}_{r,s}^{(\alpha,\psi)} + \mathcal{G}_{s,r}^{(\alpha,\psi)}) \times \sigma(r,s). \tag{73}$$

The sign $\sigma(r,s) = \det(\Pi_{\text{sort}}) = \langle\alpha|(-1)^{\sum_{i=\min(r,s)+1}^{\max(r,s)-1}\hat{n}_i}|\alpha\rangle$ takes care of the number of permutations required for sorting the columns of $P_\beta$.

This lowrank update of the ratios of Slater determinants is well-known from conventional VMC using Markov chains where is used to calculate the acceptance rate for a Monte Carlo update $|\alpha\rangle \to |\beta\rangle$. What is needed for the purposes of the algorithm presented in the main text is only the relative sign($\langle\beta/\psi\rangle/\langle\alpha/\psi\rangle$) of all "one-hop states" $|\beta\rangle$ relative to the reference state $|\alpha\rangle$.

## C  Cooptimzation of the Slater determinant

When cooptimizing the occupied orbitals of the Slater determinant together with the Jastrow factor it must be ensured that they remain orthonormal. This is done by applying an orthogonal

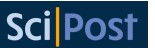

(a)

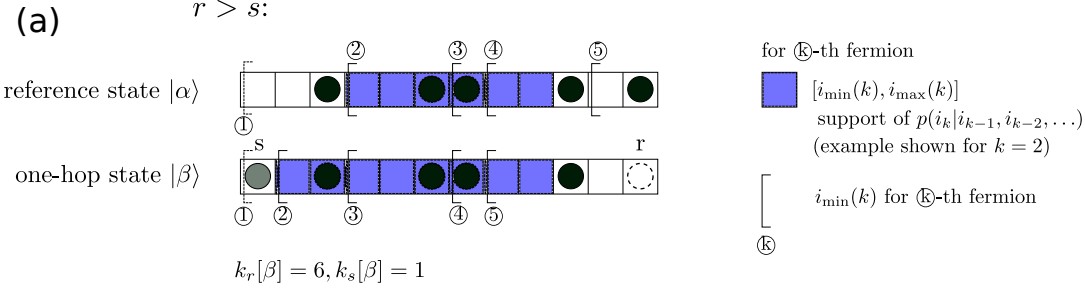

(b)

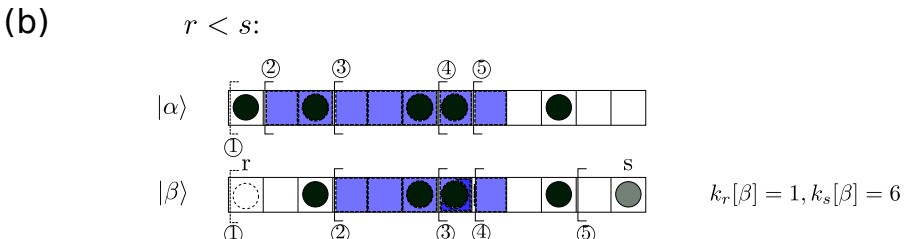

Figure 18: Illustration of the special cases arising for hopping in 1D with periodic boundary conditions. (a) For $r > s$, the support $[i_{\min}(k), i_{\max}(k)]$ (blue shaded boxes for $k = 2$) of $p_{\mathrm{cond}}^{(\beta)}(k, :)$ is larger to the left than that of $p_{\mathrm{cond}}^{(\alpha)}(k, :)$ because the particle number ordering has changed for particles $k_s[\beta] < k < k_r[\beta]$ as a result of the particle hopping from position $r$ to $s$. Therefore some additional conditional probabilities $p_{\mathrm{cond}}^{(\beta)}(k, j_{\mathrm{add}})$ for $j_{\mathrm{add}} \in \{i_{k-1}^{(\beta)} + 1, \ldots, i_{k-1}^{(\alpha)}\}$ need to be calculated which have no counterpart in the reference state $|\alpha\rangle$. (b) For $r < s$, the support of $p_{\mathrm{cond}}^{(\beta)}(k, :)$ is smaller than that of $p_{\mathrm{cond}}^{(\alpha)}(k, :)$. Again, the particle numbering has changed in state $|\beta\rangle$ due to a particle hopping from $r$ to $s$: The $k$-th particle in $|\beta\rangle$ corresponds to the $(k + 1)$-th particle in the reference state. Therefore the conditional probabilities for the $k$-th particle in state $|\beta\rangle$ are updated based on those for the $(k + 1)$-th particle in the reference state $|\alpha\rangle$ (see Fig. 19).

matrix $R$ to the matrix $U_{\mathrm{HF}}$ whose columns are the single-particle eigenstates of the Hartree-Fock Hamiltonian. Selecting the first $N_p$ columns as occupied orbitals one obtains the P-matrix representation in Eq. (64) as

$$P(R) = [R \, U_{\mathrm{HF}}]_{1:N_s, 1:N_p} \, . \tag{74}$$

For orthonormal orbitals the expression of the single-particle Green's function in Eq. (65) simplifies to

$$G(R) = \mathbb{1}_{N_s} - P(R)P(R)^T \, . \tag{75}$$

The orthogonal property of $R$ is guaranteed by writing it as the matrix exponential of a skew-symmetric matrix, specifically $R = e^{T - T^T}$, where $T$ is a strictly lower triangular matrix. The $\frac{n(n-1)}{2}$ non-zero real entries of $T$ give a non-redundant parametrization of all proper rotation matrices $R \in SO(n)$. The entries of $T$ are cooptimized together with the Jastrow factor using automatic differentiation. At the beginning of the optimization $T$ is initialized to zero so that the Hartree-Fock Slater determinant is recovered.

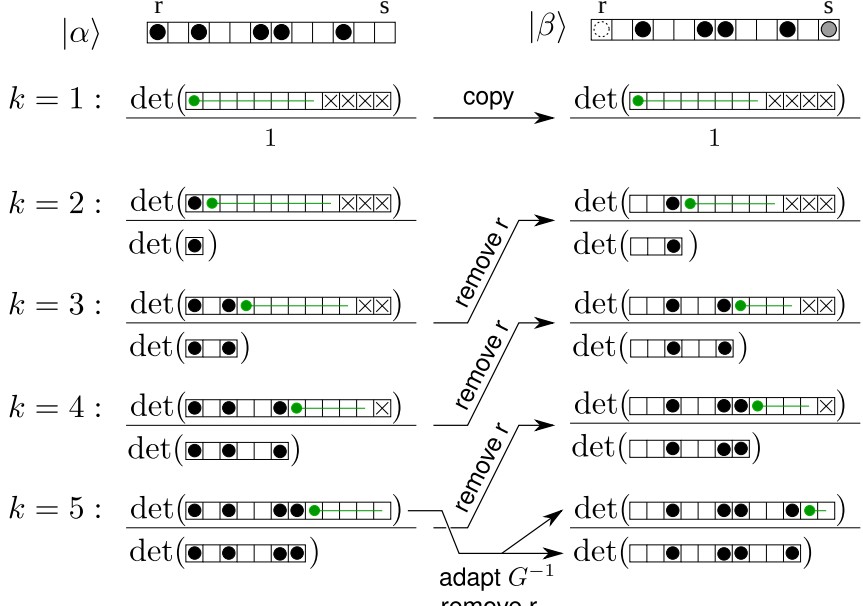

Figure 19: **Lowrank update scheme for** $r < s$ and hopping across periodic boundary conditions in 1D. $|\beta\rangle$ arises from reference state $|\alpha\rangle$ by a particle hopping across periodic boundary conditions in 1D from the first ($r$) to the last ($s$) position. Although $|\alpha\rangle$ and $|\beta\rangle$ agree in all other particle positions, the numbering has changed: Because the first particle in $|\alpha\rangle$ is missing in state $|\beta\rangle$ the $k$-th particle in $|\beta\rangle$ is the $(k+1)$-th particle in $|\alpha\rangle$. Therefore conditional probabilities for the $k$-th particle in state $|\beta\rangle$ must be calculated based on conditional probabilities for the $(k+1)$-th particle in $|\alpha\rangle$. Note that the support of the $k$-th conditional probabilities (green line) in the onehop state $|\beta\rangle$, $[i_{k-1}^{(\beta)}, N_s - (N_p - k)]$, is smaller to the left than in the reference state $|\alpha\rangle$. The dependence of the correction factors for both numerator and denominator determinants is indicated by arrows. The last case $k = N_p$ is special because numerator and denominator determinants need to be updated from the reference state at the same $k$ (rather than $k + 1$).

## D  Iterative update of the Schur complement

For a matrix

$$M = \begin{pmatrix} \tilde{X} & B \\ C & D \end{pmatrix}, \tag{76}$$

with invertible submatrix $\tilde{X}$, the Schur complement of $\tilde{X}$ in $M$ is defined as

$$S = D - C\tilde{X}^{-1}B, \tag{77}$$

and it holds that

$$\det(M) = \det(\tilde{X})\det(S), \tag{78}$$

which is the determinant formula for block matrices.

Let us consider the conditional probability for the $k$-th particle to be at two positions from the position $i_{k-1}$ of the $(k-1)$-th particle as well as the conditional probability for it to be at

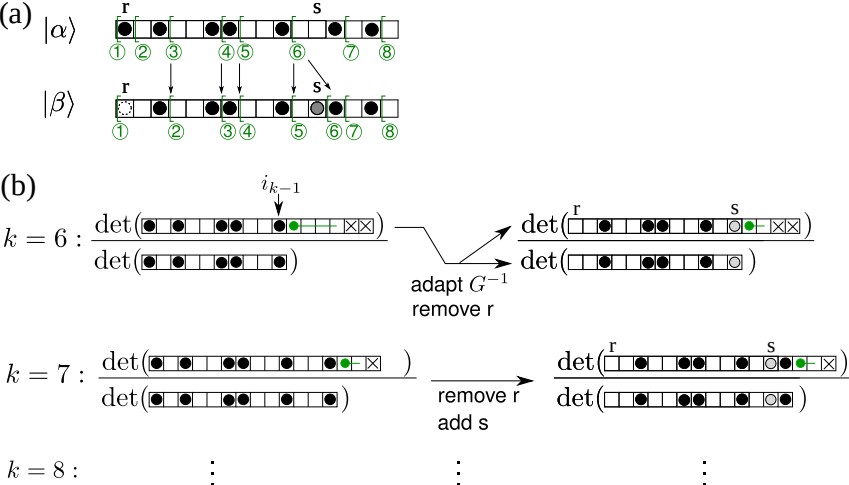

Figure 20: $r < s$: Additional special cases for hopping in 2D. (a) Green brackets with circled numbers indicate the beginning of the support for the conditional probabilities of the ⓚ-th particle. (b) Up to particle $k = k_s[\beta] = 6$ the strategy is the same as in Fig. 19. For $k > k_s[\beta]$ the lowrank update simplifies again, consisting in "removing" a particle at $r$ and "adding" a particle at $s$ both in the numerator and denominator determinants; the particle numbering is again the same in $|\alpha\rangle$ and $|\beta\rangle$ (see encircled numbers in panel (a)).

three positions from $i_{k-1}$ and carve out how the expressions change. In the former case

$$p_{\text{cond}}(k, i = i_{k-1} + 2)$$

$$= (-1)^{n_{i_{k-1}+2}} \det \left\{ \underbrace{\begin{pmatrix} G_{i_{k-1}+1,i_{k-1}+1} & G_{i_{k-1}+1,i_{k-1}+2} \\ G_{i_{k-1}+2,i_{k-1}+1} & G_{i_{k-1}+2,i_{k-1}+2} - 1 \end{pmatrix}}_{D_l} - C_l \underbrace{(G_{K,K} - N_K)^{-1}}_{\tilde{X}^{-1}} B_l \right\} \quad (79)$$

$$\equiv (-1)^{n_{i_{k-1}+2}} \det(S_l), \quad (80)$$

with

$$B_{l=2} = \begin{pmatrix} G_{1,i_{k-1}+1} & G_{1,i_{k-1}+2} \\ G_{2,i_{k-1}+1} & G_{2,i_{k-1}+2} \\ \vdots & \vdots \\ G_{i_{k-1},i_{k-1}+1} & G_{i_{k-1},i_{k-1}+2} \end{pmatrix}, \quad (81)$$

and $C_l = B_l^T$. $S_l$ and $D_l$ are $l \times l$ matrices where $l = i - i_{k-1}$ is the distance of the position $i$ under consideration from the last sampled position $i_{k-1}$. The conditional probability for placing the $k$-th particle at the next position is given by matrices which have grown by one row and one column:

$$p_{\text{cond}}(k, i = i_{k-1} + 3) = (-1)^{n_{i_{k-1}+3}} \quad (82)$$

$$\times \det \left\{ \begin{pmatrix} G_{i_{k-1}+1,i_{k-1}+1} & G_{i_{k-1}+1,i_{k-1}+2} & G_{i_{k-1}+1,i_{k-1}+3} \\ G_{i_{k-1}+2,i_{k-1}+1} & G_{i_{k-1}+2,i_{k-1}+2} & G_{i_{k-1}+2,i_{k-1}+3} \\ G_{i_{k-1}+3,i_{k-1}+1} & G_{i_{k-1}+3,i_{k-1}+2} & G_{i_{k-1}+3,i_{k-1}+3} - 1 \end{pmatrix} - C_{l+1}(G_{K,K} - N_K)^{-1} B_{l+1} \right\} \quad (83)$$

$$= (-1)^{n_{i_{k-1}+3}} \det(S_{l+1}). \quad (84)$$

Note also that the matrix element marked red has changed from $G_{i_{k-1}+2,i_{k-1}+2} - 1$ to $G_{i_{k-1}+2,i_{k-1}+2}$. Generally, the conditional probabilities for the positions of the $k$-th fermion are

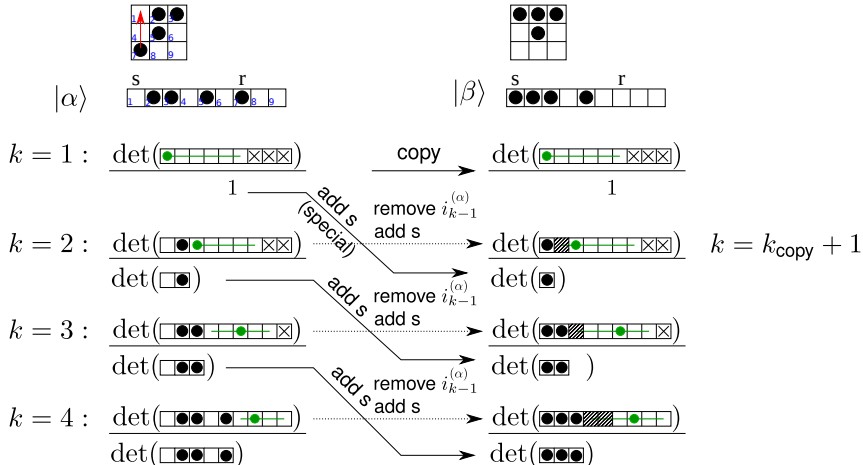

Figure 21: **Lowrank update scheme for** $r > s$. Nearest-neighbour hopping in 2D translates into long-range hopping in the 1D system of ordered positions, i.e. unlike for nearest neighbour hopping there are particles between position $s$ and $r$, and for those particles the particle number index changes by $+1$ in state $|\beta\rangle$ compared to state $|\alpha\rangle$. The position of the $k$-th particle in state $|\alpha\rangle$ is denoted as $i_k^{(\alpha)}$ and $k_s$ $(k_r)$ is the particle number index of the particle at position $s$ in state $|\beta\rangle$ (at position $r$ in state $|\alpha\rangle$). For $k \leq k_{\text{copy}}[\beta]$, the conditional probabilities can be copied identically from the reference state. In the example $k_{\text{copy}} = 1$. For $k > k_{\text{copy}}[\beta]$, the denominator determinant $\det(G_{\text{denom}}^{(\beta)})[k, i]$ is updated based on $\det(G_{\text{denom}}^{(\alpha)})[k-1, i)]$ by adding a particle at position $s$, and the numerator determinant $\det(G_{\text{num}}^{(\beta)})[k, i]$ is updated based on $\det(G_{\text{num}}^{(\alpha)})[k, i]$ by adding a particle at $s$ and removing the particle at $i_{k-1}^{(\alpha)}$ which is only there in the reference state $|\alpha\rangle$. Note that the support of the $k$-th conditional probability, indicated by a green line, in state $|\beta\rangle$, $I_k^{(\beta)} = [i_{k-1}^{(\beta)} + 1, i_{\max}]$ is larger to the left than in reference state $|\alpha\rangle$ (these additional positions are hatched in the figure). This is due to the fact that $i_{k-1}^{(\alpha)} > i_{k-1}^{(\beta)}$ since the particle number index $k$ has been shifted by $+1$ after a particle jumped from position $r$ to $s$. In other words, for $k_s[\beta] < k < k_r[\beta]$ the $k$-th particle in state $|\beta\rangle$ corresponds to the $(k-1)$-th particle in the reference state $|\alpha\rangle$.

given by the determinant of Schur complements of increasing size. Since in each step the Schur complement grows just by one row and one column, the calculation of determinants can be avoided altogether, which will be demonstrated below. Furthermore, the Schur complement, like the single-particle Green's function, is a symmetric matrix.

While calculating the conditional probabilities of the $k$-th particle, $\tilde{X}^{-1} \equiv (G_{K,K} - N_K)^{-1}$ stays constant whereas the matrices $B_l, C_l$, and $D_l$ grow. The repeated multiplications $C_l(G_{K,K} - N_K)^{-1}B_l$ and the repeated determinant evaluations are very costly. By reusing already computed results large computational savings are possible, which is illustrated schematically in Fig. 22.

Applying the formula for block determinants to the second row of Fig. 22 we obtain

$$\det(S_{l+1}) = \det(S_l')\det(S_D - S_B^T S_l'^{-1} S_B). \tag{85}$$

Note that $S_l'$ differs from $S_l$ by just by one element, the element in the lower right corner:

$$\left(S_l'\right)_{l,l} = (S_l)_{l,l} + 1. \tag{86}$$

$\det(S_l')$ can be calculated from the already known quantitiy $\det(S_l)$ by using the Sherman-



Figure 22: Graphical representation of the iterative update of the Schur complement in Eq. (30), when calculating the $l$-th conditional probability for the $k$-th particle. Blue shading in the $(l+1)$-th step indicates matrix entries that have already been used in the $l$-th step (to the left of the second equality sign) or that have already been computed - up to small modifications - in the $l$-th step (to the right of the second equality sign). The primed block matrices $D'_l$ and $S'_l$ differ from the unprimed ones only in the lower right matrix element (see main text).

Morrison formula

$$
\begin{aligned}
\det(S'_l) &= \det\left(S_l + \hat{e}_l \hat{e}_l^T\right) \\
&= \det(S_l)\det\left(1 + \hat{e}_l^T S_l^{-1} \hat{e}_l\right) = \det(S_l)(1 + \left(S_l^{-1}\right)_{l,l}).
\end{aligned}
\tag{87}
$$

Similarly, $\left(S'_l\right)^{-1}$ in Eq. (85) is obtained from $S_l^{-1}$ using a low-rank update for the inverse matrix

$$
\begin{aligned}
\left(S'_l\right)^{-1} &= S_l^{-1} - \frac{1}{1 + \left(S_l^{-1}\right)_{l,l}} S_l^{-1}
\begin{pmatrix}
0 & \cdots & & 0 \\
\vdots & \ddots & & \vdots \\
0 & \cdots & 0 & 0 \\
0 & \cdots & 0 & 1
\end{pmatrix}
S_l^{-1} \\
&= S_l^{-1} - \frac{1}{1 + \left(S_l^{-1}\right)_{l,l}} \left(S_l^{-1}\right)_{1:l,l} \otimes \left(S_l^{-1}\right)_{l,1:l}.
\end{aligned}
\tag{88}
$$

Likewise, since $(S_l)_{l,l} = (S'_l)_{l,l} - 1$

$$
S_l^{-1} = S_l'^{-1} + \frac{1}{1 - \left(S_l'^{-1}\right)_{l,l}} \left(S_l'^{-1}\right)_{1:l,l} \otimes \left(S_l'^{-1}\right)_{l,1:l}.
\tag{89}
$$

In summary

$$
\det(S_{l+1}) = \det(S_l)(1 + \left(S_l^{-1}\right)_{l,l})\det(S_D - S_B^T (S'_l)^{-1} S_B),
\tag{90}
$$

with $S_l'^{-1}$ given by Eq. (88). Compared to the direct evaluation of the determinant on the left-hand side of Eq. (90), which costs $(l+1)^3$ operations, the vector-matrix-vector product $S_B^T (S'_l)^{-1} S_B$ on the right-hand side requires only $l^2 + l$ operations. $(S_l^{-1})_{l,l}$ is calculated from $S_l'^{-1}$ through

$$
\left(S_l^{-1}\right)_{l,l} = \left(S_l'^{-1}\right)_{l,l} + \frac{\left(S_l'^{-1}\right)_{l,l}\left(S_l'^{-1}\right)_{l,l}}{1 - \left(S_l'^{-1}\right)_{l,l}} = \frac{\left(S_l'^{-1}\right)_{l,l}}{1 - \left(S_l'^{-1}\right)_{l,l}}.
\tag{91}
$$

Given $S'^{-1}_{l-1}$ one can calculate $S'^{-1}_l$ using the formula for the inverse of a block matrix.

# E   Lowrank updates for local kinetic energy

Supplementing Sec. 2.5, the code listing 1 specifies how to obtain the conditional probabilities $p^{(\beta)}_{\text{cond}}(k,i)$ for all "onehop" states $|\beta\rangle$ connected by single-particle hopping to a common reference state $|\alpha\rangle$ from $p^{(\alpha)}_{\text{cond}}(k,i)$ using a set of low-rank updates.

---

**Algorithm 1** Pseudocode of lowrank update for local kinetic energy

---

**Input:** Conditional probabilities $p^{(\alpha)}_{\text{cond}}(k,i)$ in reference state $|\alpha\rangle$ for particle number $k \in \{1,\dots,N_p\}$ and position $i \in I^{(\alpha)}_k$

**Input:** Onehop states $|\beta\rangle$ ordered according to increasing $k_{\text{copy}}[\beta]$ for $\beta = 1,\dots,N_{\text{onehop states}}$

**Input:** $r[\beta], s[\beta]$: onehop state $|\beta\rangle$ is generated from reference state $|\alpha\rangle$ by letting a particle hop from position $r[\beta]$ to $s[\beta]$

**Input:** $k_r[\beta], k_s[\beta]$: numbering of particles at positions $r[\beta], s[\beta]$, as counted in onehop state $|\beta\rangle \rightarrow$ see Fig. 18(b)

**Input:** $i^{(\alpha)}_k, i^{(\beta)}_k$: position of the $k$-th particle in state $|\alpha\rangle, |\beta\rangle$

**Output:** Conditional probabilities $p^{(\beta)}_{\text{cond}}(k,i)$ for all $\beta = 1,\dots,N_{\text{onehop states}}$

---

```
 1: for each particle k = 1, N_p do                                     ▷ k refers to particle numbering in |α⟩
 2:     I_k^(α) = [i_{k-1}^(α) + 1, N_s - (N_p - k)]                     ▷ I_k^(α) is support of p_cond^(α)(k,:)
 3:     for each ordered position i ∈ I_k^(α) do
 4:         for each "onehop state" β = 1, N_onehop states do
 5:             if k ≤ k_copy[β] then
 6:                 p_cond^(β)(k,:) ← p_cond^(α)(k,:)                    ▷ up to k_copy[β] conditional probabilities are identical
 7:             else
 8:                 if r[β] < s[β] then                                 ▷ → see Fig. 18(b) and Fig. 19
 9:                     if k > 1 and k ≤ k_s[β] then
10:                         κ^(r)(i) ← Eqs. 92                          ▷ correction factor κ, remove-r
11:                         p_cond^(β)(k-1,i) = κ^(r)(i) × p_cond^(α)(k,i)
12:                         if k = k_s[β] or k = N_p then               ▷ → see Fig. 20
13:                             p_cond^(β)(k,i) ← Eq.(95l) for i ∈ [i_{k-1}^(β), N_s - (N_p - k)]
14:                         end if
15:                     else if k = k_s[β] + 1 and i > s[β] then
16:                         κ ← Eq.(96i)
17:                         p_cond^(β)(k,i) = κ × p_cond^(α)(k,i)
18:                     else if k > k_s[β] + 1 and i > s[β] then
19:                         κ^(r,s)(i) ← Eq. (93)                       ▷ remove-r-add-s
20:                         p_cond^(β)(k,i) = κ^(r,s)(i) × p_cond^(α)(k,i)
21:                     end if
22:                 else if r[β] > s[β] then                            ▷ → see Fig. 21
23:                     if k ≤ k_r[β] then
24:                         I_k^(β) = {i_{k-1}^(β) + 1,...,i_{k-1}^(α)} ∪ I_k^(α)
25:                         if i = i_{k-1}^(α) then
26:                             Calculate additionally ...               ▷ ...since I_k^(β) is larger than I_k^(α)
27:                             for j_add ∈ {i_{k-1}^(β) + 1,...,i_{k-1}^(α)} do     ▷ see Fig. 18(a)
28:                                 κ_denom ← Eq. (97g)                  ▷ reuse G_denom^(α)-1[k-1] and G_num^(α)-1[k-1, j_add]
29:                                 κ_num ← κ^(s)(j_add) Eq. (97a)                   ▷ add-s
30:                                 p_cond^(β)(k,j_add) = (κ_num/κ_denom) × p_cond^(α)(k-1,j_add)
31:                             end for
32:                         end if
33:                         κ_num ← κ^(i_{k-1}^(α),s[β])(i) Eq. (97i)             ▷ remove-r-add-s in numerator
34:                         if k = k_copy[β] + 1 then
35:                             κ_denom ← Eqs.(97f), (97g)               ▷ extend-Gdenominv
36:                         else
37:                             κ_denom ← κ^(s[β]) Eq. (97j)              ▷ add-s
38:                         end if
```

39: $\quad p_{\text{cond}}^{(\beta)}(k,i) = \frac{\kappa_{\text{num}}}{\kappa_{\text{denom}}} \times \frac{\det(G_{\text{denom}}^{(\alpha)}[k])}{\det(G_{\text{denom}}^{(\alpha)}[k-1])} \times p_{\text{cond}}^{(\alpha)}(k,i)$ Eq. (97k)

40: $\quad\quad$ **else if** $k > k_r[\beta]$ **then**

41: $\quad\quad\quad \kappa^{(r,s)} \leftarrow$ Eq. (93) $\quad\quad\quad\quad\quad\quad\quad\quad\quad\quad\quad\quad\quad\quad$ ▷ **remove-r-add-s**

42: $\quad\quad\quad p_{\text{cond}}^{(\beta)}(k,i) = \kappa^{(r,s)}(i) \times p_{\text{cond}}^{(\alpha)}(k,i)$

43: $\quad\quad$ **end if**

44: $\quad\quad$ **end if**

45: $\quad$ **end if**

46: $\quad$ **end for**

47: $\quad$ **end for**

48: **end for**

## E.1 remove-r update

The **remove-r** update is illustrated in Fig. 23. Using the lowrank update of the determinant given by Eqs. (49) and (51) with $U^{(r)} = V^{(r)} = \hat{e}_r$, the total correction factor for the **remove-r** adjustment shown in Fig. 23 becomes

$$
\kappa^{(r)}(i) = \textbf{remove\_r}\Big( G_{\text{num}}^{(\alpha)-1}[k,i], G_{\text{denom}}^{(\alpha)-1}[k] \Big)
$$
$$
= \frac{1 + \big( G_{\text{num}}^{(\alpha)-1}[k,i] \big)_{r,r}}{1 + \big( G_{\text{denom}}^{(\alpha)-1}[k] \big)_{r,r}}, \tag{92a}
$$

with the numerator and denominator matrices

$$
G_{\text{num}}^{(\alpha)}[k,i] = G_{1:i,1:i} - N_{1:i}^{(\alpha)} \tag{92b}
$$
$$
G_{\text{denom}}^{(\alpha)}[k] = G_{1:i_{k-1}^{(\alpha)},1:i_{k-1}^{(\alpha)}} - N_{1:i_{k-1}^{(\alpha)}}^{(\alpha)}, \tag{92c}
$$

whose inverses are assumed to be known from the processing of state $|\alpha\rangle$.

## E.2 remove-r-add-s update

The **remove-r-add-s** update is shown in Fig. 24. As derived in the main text in Eq. (54), the total correction factor for removing a particle at $r$ and adding a particle at $s$, both in the numerator and denominator determinant, is as follows

$$
\kappa^{(r,s)}(i) = \frac{(1 + (G_{\text{num}}^{(\alpha)-1}[k,i])_{r,r})(1 - (G_{\text{num}}^{(\alpha)-1}[k,i])_{s,s}) + (G_{\text{num}}^{(\alpha)-1}[k,i])_{r,s}(G_{\text{num}}^{(\alpha)-1}[k,i])_{s,r}}{(1 + (G_{\text{denom}}^{(\alpha)-1}[k])_{r,r})(1 - (G_{\text{denom}}^{(\alpha)-1}[k])_{s,s}) + (G_{\text{denom}}^{(\alpha)-1}[k])_{r,s}(G_{\text{denom}}^{(\alpha)-1}[k])_{s,r}}, \tag{93}
$$

with the matrices from Eqs. (92b),(92c).

## E.3 extend-Gnum-remove-r update

The **extend-Gnum-remove-r** update is used for $r[\beta] < s[\beta], k = k_s[\beta], i > i_{k-1}^{(\beta)}$; see Fig. 25. First notice that the inverse of the matrix

$$
G_{\text{num}}^{(\alpha)}[k, i_{k-1}^{(\beta)}] = G_{1:i_{k-1}^{(\beta)},1:i_{k-1}^{(\beta)}} - N_{1:i_{k-1}^{(\alpha)}}^{(\alpha)} \cup \{0,0,\ldots,0,1_{i_{k-1}^{(\beta)}}\}, \tag{94}
$$

$$
p_{\text{cond}}^{(\alpha)}(k=3,i) = \frac{\det(\stackrel{r}{\boxed{\bullet}}\,\boxed{\bullet}\,\boxed{+}\boxed{+}\,\boxed{\bullet}\,\stackrel{s}{\boxed{\times}})}{\det(\boxed{\bullet}\,\boxed{\bullet})} \xrightarrow[\text{remove } r]{\text{remove } r} \frac{\det(\stackrel{r}{\boxed{\,}}\,\overset{i_{k-1}^{(\alpha)}}{\boxed{\bullet}}\,\boxed{+}\boxed{+}\,\overset{i}{\boxed{\bullet}}\,\stackrel{s}{\boxed{\times}}\boxed{\times})}{\det(\boxed{\,}\,\boxed{\bullet})} = p_{\text{cond}}^{(\beta)}(k=2,i)
$$
$$
= \kappa^{(r)}(i) \times p_{\text{cond}}^{(\alpha)}(k=3,i)
$$

Figure 23: **remove-r** adjustment, see Eqs. (92). Example for four particles on nine sites.

$$p_{\text{cond}}^{(\alpha)}(k=3,i) = \frac{\det(\bullet\ \bullet\ \boxed{\cdots}\ \bullet\ \boxtimes)}{\det(\bullet\ \bullet)} \xrightarrow[\substack{\text{remove } r \\ \text{add } s}]{\substack{\text{remove } r \\ \text{add } s}} \frac{\det(\bullet\bullet\ \boxed{\cdots}\ \bullet\ \boxtimes)}{\det(\bullet\bullet)} = p_{\text{cond}}^{(\beta)}(k=3,i)$$
$$= \kappa^{(r,s)}(i) \times p_{\text{cond}}^{(\alpha)}(k=3,i)$$

Figure 24: **remove-r-add-s** adjustment, see Eqs. (93),(92b),(92c).

$$p_{\text{cond}}^{(\alpha)}(k=4,i) = \frac{\det(\bullet\ \bullet\bullet\ \boxed{\cdots})}{\det(\bullet\ \bullet\bullet)} \longrightarrow \frac{\det(\ \bullet\bullet\ \bullet\ \bullet)}{\det(\ \bullet\bullet\ \bullet)} = p_{\text{cond}}^{(\beta)}(k=k_s[\beta],i)$$
$$= \frac{\kappa_1(i)\kappa_2(i)}{\kappa_3}$$

Figure 25: **extend-Gnum-remove-r**: The meaning of the two arrows in the upper part of the figure is explained in the lower part: Extend $G_{\text{num}}^{(\alpha)-1}[k,i_{k-1}^{(\beta)}]$ from $i_{k-1}^{(\beta)}+1$ to $i$ (extended positions shaded grey) via block update of the inverse matrix according to Eq. (95i).

where the last 1 in $\{0,0,\ldots,0,1_{i_{k-1}^{(\beta)}}\}$ is at position $i_{k-1}^{(\beta)}$, is assumed to have been computed for the reference state $|\alpha\rangle$. (Remark: One does not need to compute the inverse of $G_{\text{num}}^{(\alpha)}[k,i_{k-1}^{(\beta)}]$ each time from scratch, but one can update it iteratively from some previously computed $G_{\text{num}}^{(\alpha)-1}[k',i']$ for some $k' \le k$ and $i' < i_{k-1}^{(\beta)}$ using a block update.) The block structure of the extended numerator matrix is

$$A = G_{\text{num}}^{(\alpha)}[k,i_{k-1}^{(\beta)}] = G_{1:i_{k-1}^{(\beta)},1:i_{k-1}^{(\beta)}} - N_{1:i_{k-1}^{(\alpha)}}^{(\alpha)} \cup \{0,0,\ldots,0,1\} \quad \to A^{-1} \text{ is known}, \tag{95a}$$

$$B = G_{1:i_{k-1}^{(\beta)},i_{k-1}^{(\beta)}+1:i}, \tag{95b}$$

$$C = B^T, \tag{95c}$$

$$D = G_{i_{k-1}^{(\beta)}+1:i,i_{k-1}^{(\beta)}+1:i} - \text{diag}(0,0,\ldots,0,1_i), \tag{95d}$$

and

$$\det(G_{\text{num, extended}}) = \det\begin{pmatrix} A & B \\ C & D \end{pmatrix} = \det(A)\underbrace{\det(D-CA^{-1}B)}_{\kappa_1(i)} \tag{95e}$$

$$= \kappa_1(i)\det(G_{\text{num}}^{(\alpha)}[k,i_{k-1}^{(\beta)}]), \tag{95f}$$

where the determinant of the Schur complement of $A$, that is $S = D - CA^{-1}B$, has been marked as an intermediate correction factor $\kappa_1(i)$. To obtain the numerator determinant of the onehop state $|\beta\rangle$, a particle needs to be removed at position $r$ from $G_{\text{num, extended}}$, that is

$$\left(G_{\text{num, extended}}\right)_{r,r} \to \left(G_{\text{num, extended}}\right)_{r,r} + 1. \tag{95g}$$

This results in another intermediate correction factor to the numerator determinant of the onehop state $|\beta\rangle$:

$$\kappa_2(i) = 1 + \left(G_{\text{num, extended}}^{-1}\right)_{r,r}, \tag{95h}$$

where the inverse of the extended numerator matrix has been obtained via a block update:

$$G_{\text{num, extended}}^{-1} = \begin{pmatrix} A & B \\ C & D \end{pmatrix}^{-1} = \begin{pmatrix} A^{-1} + A^{-1}BS^{-1}CA^{-1} & -A^{-1}BS^{-1} \\ -S^{-1}CA^{-1} & S^{-1} \end{pmatrix}. \tag{95i}$$

Thus, for $i > i_{k-1}^{(\beta)}$

$$\det(G_{\text{num}}^{(\beta)}[k = k_s[\beta], i]) = \kappa_2(i) \times \kappa_1(i) \times \det(G_{\text{num}}^{(\alpha)}[k, i_{k-1}^{(\beta)}]). \tag{95j}$$

The denominator determinant is obtained directly from the determinant of $G_{\text{num}}^{(\alpha)}[k, i_{k-1}^{(\beta)}]$ by removing a particle at $r$

$$\det(G_{\text{denom}}^{(\beta)}[k = k_s[\beta]]) = \underbrace{\left(1 + \left(G_{\text{num}}^{(\alpha)}[k, i_{k-1}^{(\beta)}]\right)_{r,r}\right)}_{=\kappa_3} \times \det(G_{\text{num}}^{(\alpha)}[k, i_{k-1}^{(\beta)}]). \tag{95k}$$

Collecting the correction factors for numerator and denominator determinants one obtains for the conditional probability in the onehop state:

$$p_{\text{cond}}^{(\beta)}(k = k_s[\beta], i) = \frac{\det(G_{\text{num}}^{(\beta)}[k=k_s[\beta], i])}{\det(G_{\text{denom}}^{(\beta)}[k])} = \frac{\kappa_1(i)\kappa_2(i) \times \cancel{\det(G_{\text{num}}^{(\alpha)}[k, i_{k-1}^{(\beta)}])}}{\kappa_3 \times \cancel{\det(G_{\text{num}}^{(\alpha)}[k, i_{k-1}^{(\beta)}])}} = \frac{\kappa_1(i)\kappa_2(i)}{\kappa_3}. \tag{95l}$$

One may wonder whether the use of Eq. (95l) gives any efficiency gain compared to the direct calculation of the determinant ratio. The key point is that the matrices $B, C$, and $S$ in eq. (95i) are of dimension $i_{k-1}^{(\beta)} \times (i - i_{k-1}^{(\beta)})$ and $(i - i_{k-1}^{(\beta)}) \times (i - i_{k-1}^{(\beta)})$, respectively, and on average $(i - i_{k-1}^{(\beta)}) \ll i$, so that the calculation of the block inverse Eq. (95i) is less expensive than the calculation of a determinant of the $i \times i$ matrix $G_{\text{num}}^{(\beta)}[k, i]$.

### E.4  extend-Gdenom-remove-r-add-s update

The **extend-Gdenom-remove-r-add-s** is used for $r[\beta] < s[\beta], k = k_s[\beta] + 1, i > s[\beta]$; see Fig. 26. To obtain the numerator matrix of the $|\beta\rangle$ state from that of the $|\alpha\rangle$ state one needs to remove a particle at position $r$ and put a particle at position $s$, and consequently for the numerator determinant there is a correction factor

$$\kappa_{\text{num}}^{(r,s)}(i) = (1 + (G_{\text{num}}^{(\alpha)-1}[k, i])_{r,r})(1 - (G_{\text{num}}^{(\alpha)-1}[k, i])_{s,s}) + (G_{\text{num}}^{(\alpha)-1}[k, i])_{r,s}(G_{\text{num}}^{(\alpha)-1}[k, i])_{s,r}. \tag{96a}$$

The denominator matrix of the $|\beta\rangle$ state is obtained by extending the (inverse of the) denominator matrix of the $|\alpha\rangle$ state and then removing a particle at position $r$. This results in two correction factors, one for the block update of the inverse denominator matrix according to Eq. (95e)

$$\kappa_1 = \det(D - CA^{-1}B), \tag{96b}$$

$$p_{\text{cond}}^{(\alpha)}(k=5,i) = \frac{\det\left(\begin{array}{c}\end{array}\right)}{\det\left(\begin{array}{c}\end{array}\right)} \xrightarrow[\text{extend } G_{\text{denom}}^{-1} \\ \text{remove } r]{\overset{\text{remove } r}{\underset{\text{add } s}{\phantom{xxx}}}} \frac{\det\left(\begin{array}{c}\end{array}\right)}{\det\left(\begin{array}{c}\end{array}\right)} = \begin{array}{l} p_{\text{cond}}^{(\beta)}(k = k_s[\beta] + 1, i) \\ = \frac{\kappa^{(r,s)}(i)}{\kappa_1\kappa_2} \times p_{\text{cond}}^{(\alpha)}[k, i] \end{array}$$

Figure 26: **extend-Gdenom-remove-r-add-s**

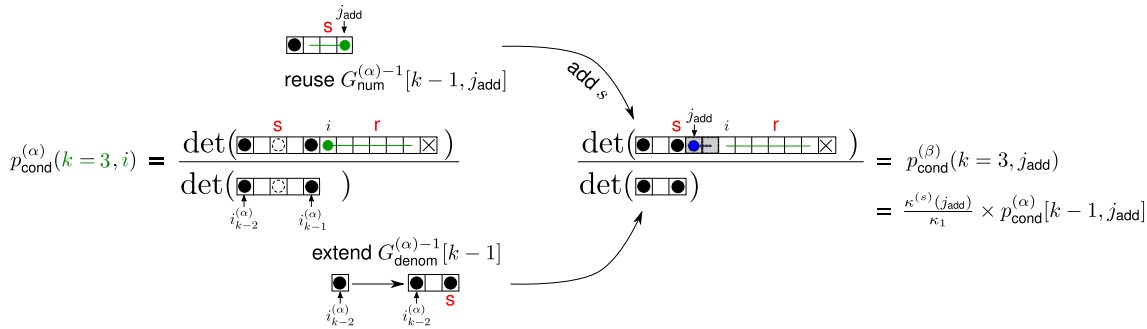

Figure 27: Since the support of the conditional probabilties in the $|\beta\rangle$ state, $I_k^{(\beta)}$, is larger to the left than in the $|\alpha\rangle$ state, $I_k^{(\alpha)}$, some additional conditional probabilities need to be calculated which have no counterpart in the $|\alpha\rangle$ state (shaded in grey). Example for $N_p = 4$ particles on $N_s = 12$ sites.

where

$$A = G_{\mathrm{denom}}^{(\alpha)}[k] = G_{1:i_{k-1}^{(\alpha)}, 1:i_{k-1}^{(\alpha)}} - N_{1:i_{k-1}^{(\alpha)}}^{(\alpha)} \, , \tag{96c}$$

$$B = G_{1:i_{k-1}^{(\alpha)}, i_{k-1}^{(\alpha)}+1:s} \, , \tag{96d}$$

$$C = B^T \, , \tag{96e}$$

$$D = G_{i_{k-1}^{(\alpha)}+1:s, i_{k-1}^{(\alpha)}+1:s} - \mathrm{diag}(0, 0, \ldots, 0, 1_s) \quad \to \text{put a particle at position } s \tag{96f}$$

are the block matrices in

$$G_{\mathrm{denom, \, extended}} = \begin{pmatrix} A & B \\ C & D \end{pmatrix}, \tag{96g}$$

and secondly for the removal of a particle at position $r$ using Eq. (95i) for the block update of the inverse of te extended denominator matrix and then applying Eq. (95h) to the resulting matrix:

$$\kappa_2 = 1 + \left( G_{\mathrm{denom, \, extended}}^{-1} \right)_{r,r} \, . \tag{96h}$$

Combining the correction factors for numerator and denominator determinants one obtains the overall correction factor

$$\kappa = \frac{\kappa_{\mathrm{num}}^{(r,s)}}{\kappa_1 \kappa_2} \, . \tag{96i}$$

### E.5  Gdenom-from-Gdenom[k-1] update

The **Gdenom-from-Gdenom[k-1] update** is used for $r[\beta] > s[\beta], k = k_{\mathrm{copy}}[\beta] + 1$; see Fig. 27 as well as Fig. 21. The correction factor to the numerator determinant is

$$\kappa_{\mathrm{num}} \equiv \kappa^{(s)}(j_{\mathrm{add}}) = 1 - \left( G_{\mathrm{num}}^{(\alpha)-1}[k-1, j_{\mathrm{add}}] \right)_{s,s} \, , \tag{97a}$$

with

$$G_{\mathrm{num}}^{(\alpha)-1}[k-1, j_{\mathrm{add}}] = G_{1:i_{k-2}^{(\alpha)}, 1:i_{k-2}^{(\alpha)}} - N_{1:i_{k-2}^{(\alpha)}}^{(\alpha)} \cup \{0, \ldots, 0, 1_{j_{\mathrm{add}}}\} \, , \tag{97b}$$

whose inverse should have been calculated and stored while processing state $|\alpha\rangle$. The inverse of the denominator matrix is extended via a block update with

$$A = G_{\text{denom}}^{(\alpha)}[k-1] = G_{1:i_{k-2}^{(\alpha)}, 1:i_{k-2}^{(\alpha)}} - N_{1:i_{k-2}^{(\alpha)}}^{(\alpha)}, \tag{97c}$$

$$B = G_{1:i_{k-2}^{(\alpha)}, i_{k-2}^{(\alpha)}+1:s}, \tag{97d}$$

$$C = B^T, \tag{97e}$$

$$D = G_{i_{k-2}^{(\alpha)}+1:s, i_{k-2}^{(\alpha)}+1:s} - \text{diag}(0, 0, \ldots, 0, 1_s) \quad \rightarrow \text{put a particle at position } s, \tag{97f}$$

which results in a correction factor

$$\kappa_{\text{denom}} \equiv \kappa_1 = \det(D - CA^{-1}B) \tag{97g}$$

to the denominator determinant. Note that there is no additional correction factor for adding a particle at $s$ in the denominator because this has already been taken care of when extending the denominator inverse in Eq. (97f). In total,

$$p_{\text{cond}}^{(\beta)}[k, j_{\text{add}}] = \frac{\kappa_{\text{num}}}{\kappa_{\text{denom}}} \times p_{\text{cond}}^{(\alpha)}[k-1, j_{\text{add}}], \tag{97h}$$

i.e. for the sites $j_{\text{add}} \in \{s+1, \ldots, i_{k-1}^{(\alpha)}\}$ the lowrank update has to be based on $p_{\text{cond}}^{(\alpha)}$ for the previous particle $k-1$ because $p_{\text{cond}}^{(\alpha)}[k, :]$ has no support there.

Still referring to Fig. 27, for $i > i_{k-1}^{(\alpha)}$ (and, as before, $r[\beta] > s[\beta], k_s[\beta] < k \leq k_r[\beta]$) the numerator determinant in the $|\beta\rangle$ state is obtained from that of the $|\alpha\rangle$ state by removing the particle at $i_{k-1}^{(\alpha)}$ and adding it at position $s$, i.e. the correction factor to the numerator determinant is given by Eq. (53) of the main text (**remove-r-add-s**) with the replacement $r \leftarrow i_{k-1}^{(\alpha)}$ and $s = s[\beta]$

$$\kappa_{\text{num}}^{(r=i_{k-1}^{(\alpha)}, s)} = (1 + (G_{\text{num}}^{(\alpha)-1}[k, i])_{i_{k-1}^{(\alpha)}, i_{k-1}^{(\alpha)}})(1 - (G_{\text{num}}^{(\alpha)-1}[k, i])_{s,s}) + (G_{\text{num}}^{(\alpha)-1}[k, i])_{i_{k-1}^{(\alpha)}, s}(G_{\text{num}}^{(\alpha)-1}[k, i])_{s, i_{k-1}^{(\alpha)}}. \tag{97i}$$

The correction factor to the denominator determinant is given by Eqs. (97f) and Eqs. (97c) to (97g) for $k = k_s[\beta] + 1 \equiv k_{\text{copy}} + 1$, and for $k > k_s[\beta] + 1$ by

$$\kappa_{\text{denom}} = 1 - \left(G_{\text{denom}}^{(\alpha)-1}[k-1]\right)_{s,s}. \tag{97j}$$

Note that in either case the starting point are denominator matrices of the reference state $|\alpha\rangle$ at the previous sampling state $k-1$. Putting everything together:

$$p_{\text{cond}}^{(\beta)}[k, i] = \frac{\kappa_{\text{num}} \cdot \det(G_{\text{num}}^{(\alpha)}[k, i])}{\kappa_{\text{denom}} \cdot \det(G_{\text{denom}}^{(\alpha)}[k-1])}$$

$$= \frac{\kappa_{\text{num}}}{\kappa_{\text{denom}}} \times \frac{\det(G_{\text{denom}}^{(\alpha)}[k])}{\det(G_{\text{denom}}^{(\alpha)}[k-1])} \times \underbrace{\frac{\det(G_{\text{num}}^{(\alpha)}[k, i])}{\det(G_{\text{denom}}^{(\alpha)}[k, i])}}_{p_{\text{cond}}^{(\alpha)}[k, i]}. \tag{97k}$$

## F  Illustrative example of conditional probabilities

Figs. 28 to 30 illustrate the degree of similarity between the conditional probabilities of a reference configuration $|\alpha\rangle$ and three randomly selected "onehop"-states $|\beta_{80}\rangle, |\beta_{90}\rangle$ and $|\beta_{100}\rangle$ for a large system of 36 fermions on a square system of 144 sites (considering only the contribution from the Slater determinant).

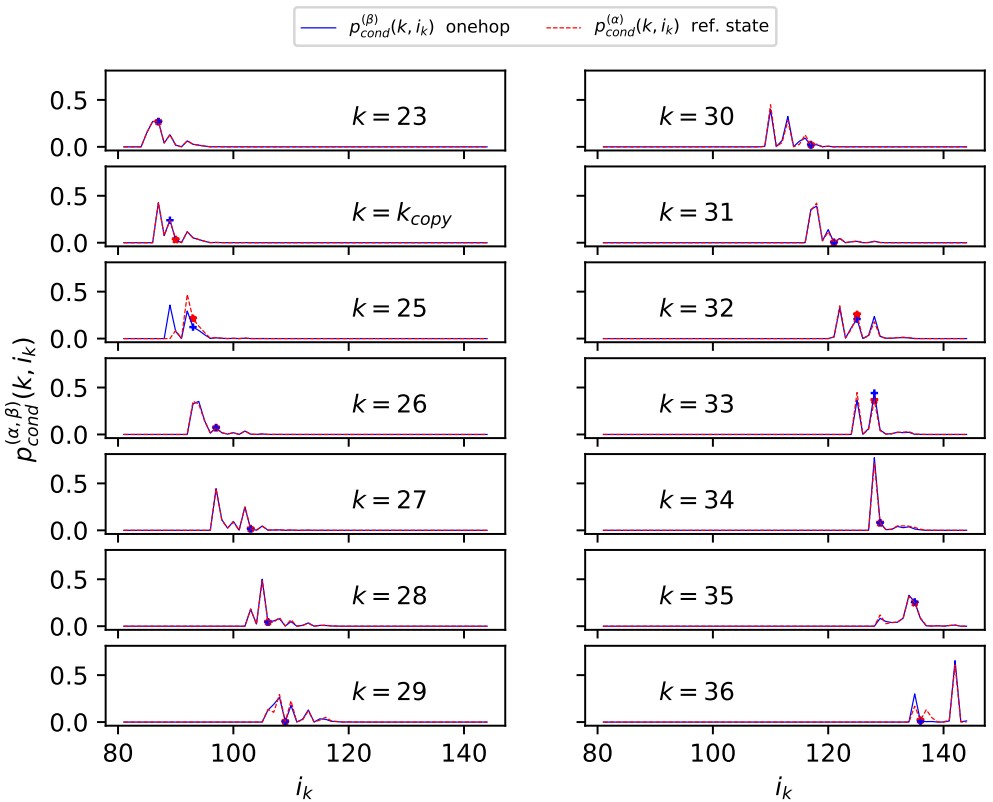

Figure 28: Conditional Slater determinant probabilities $p_{\text{cond}}^{(\alpha)}(k, i_k)$ and $p_{\text{cond}}^{(\beta)}(k, i_k)$ for a reference state $|\alpha\rangle$ and an arbitrary "onehop"-state $|\beta_{80}\rangle$ (out of 134 "onehop" states). $k$ is the ordered particle index and $i_k$ is its position. Up to $k = k_{\text{copy}}(\beta)$ the conditional probabilities coincide completely: $p_{\text{cond}}^{(\beta)}(k, i_k) = p_{\text{cond}}^{(\alpha)}(k, i_k)$ for $k \leq k_{\text{copy}}(\beta)$. "Onehop" states $|\beta\rangle$ are ordered according to increasing value of $k_{\text{copy}}(\beta)$, i.e. $k_{\text{copy}}(\beta_2) \geq k_{\text{copy}}(\beta_1)$ if $\beta_2 > \beta_1$. Red and blue dots indicate the conditional probabilities at the actually sampled positions in state $|\alpha\rangle$ and $|\beta\rangle$, respectively. Parameters: $N_x = N_y = 12, N_p = 36, V/t = 6$.

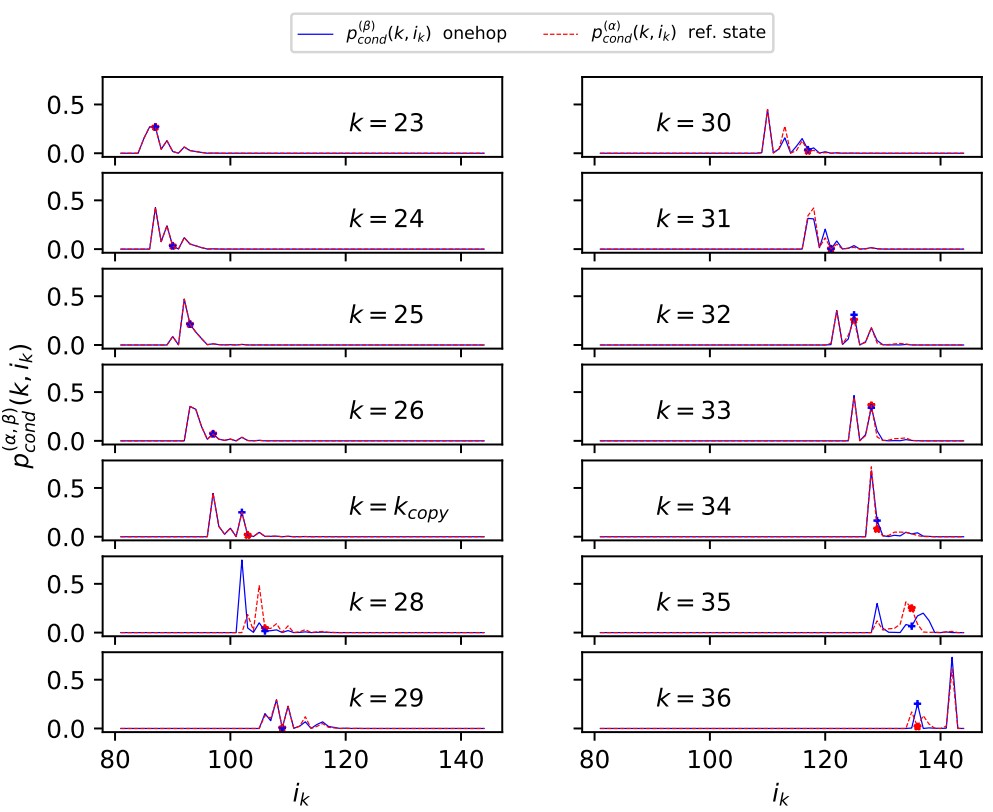

Figure 29: Conditional Slater determinant probabilities for a reference state $|\alpha\rangle$ and an arbitrary "onehop"-state $|\beta_{90}\rangle$. See Fig. 28 for more details.

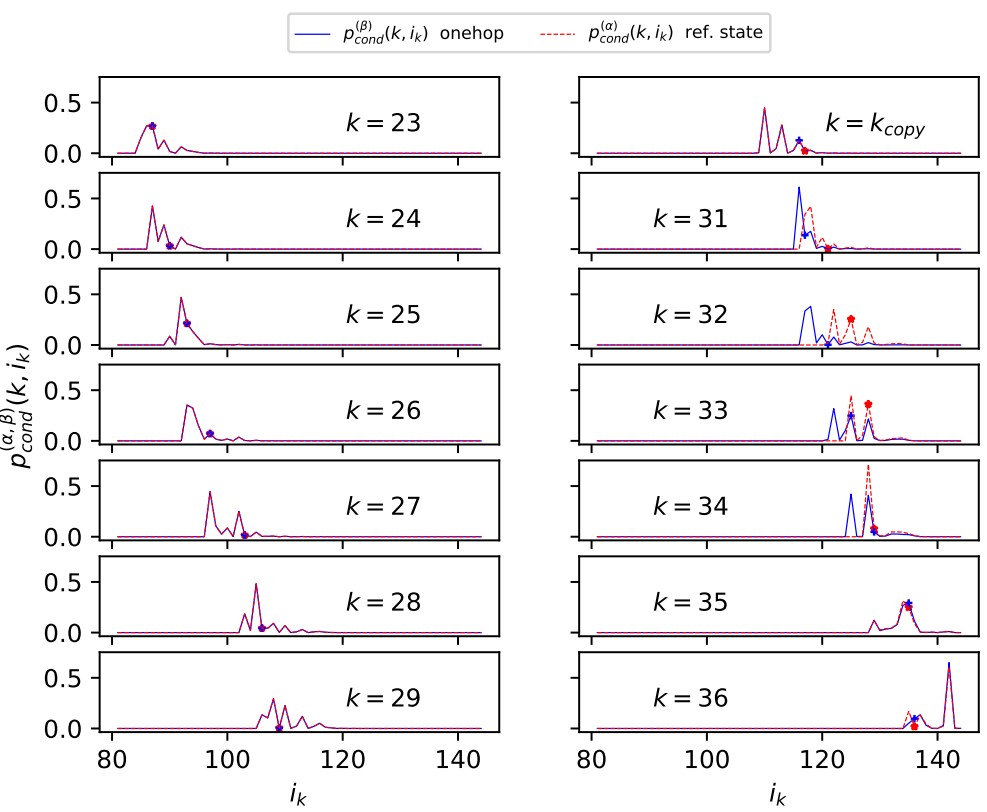

Figure 30: Conditional Slater determinant probabilities for a reference state $|\alpha\rangle$ and an arbitrary "onehop"-state $|\beta_{100}\rangle$. See Fig. 28 for more details.

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
