# Peer review of "Autoregressive neural Slater-Jastrow ansatz for variational Monte Carlo simulation"

_SciPost Physics, doi:SciPost Phys. 14, 171 (2023)_

## Round 1 · Referee Report · Anonymous · 2022-12-21

Strengths
- Detailed discussion of the algorithm
- Solid numerical results
Weaknesses
- Not easy to read
- Rather brief on the discussion of the benchmark
Report
Artificial neural networks have recently received significant attention in the context of variational Monte-Carlo (VMC) simulations. Fro example, it has been demonstrated that certain variational states based on artificial neural networks quantum sattes can be used to approximate ground states and efficient sampling techniques have been devised. In this manuscript, the authors introduce an autoregressive Slater-Jastrow ansatz. This ansatz allows for uncorrelated sampling while retaining the cubic scaling of the computational cost with system size.
The manuscript is fairly clearly written and it provides a large of amount of details regarding the implementation of the method. Given the high level of detail, it might be useful to have a short section providing a high level overview of the algorithm first.
In addition, there are a number of comments which the authors could consider:
* While there is a lot of detail regarding the method, the benchmark section is fairly short. It could be helpful to also provide the results based on other VMC methods.
* Is it possible to estimate the error density depend on the systems size?
* What are the dashed and dotted lines in Fig. 15? It would be useful to mention it in the caption (as dine in Fig. 14).
* The conclusion is rather brief. It might be useful to comment on the conclusions drawn from the benchmark and comparison to other variational methods.
* Several formulas extend beyond the column—which make it hard to read.
Overall, the manuscript represents a solid numerical work on a new approach to VMC. However, it might be useful to make it more accessible and to extend the discussion of the benchmark. I thus recommend publication following a revision.
Author: Stephan Humeniuk on 2023-03-14 [id 3473]
(in reply to Report 1 on 2022-12-21)
We thank the referee for their time and effort to review our manuscript. Please find our response to their comments below.
The referee asks to provide a high-level overview of the method first. We believe that Figs. 1 and 3 and the first page of the Method section already provide a general and high-level introduction to the algorithm. While they could be combined into a separate paragraph, this would lead to repetitions, which is why we choose to keep the previous structure.
While there is a lot of detail regarding the method, the benchmark section is fairly short. It could be helpful to also provide the results based on other VMC methods.
We have extended the benchmark section and included a comparison with increasingly accurate variants of RBM-Slater-Jastrow (SJ) wavefunctions taken from the comprehensive variational benchmark Varbench. In terms of accuracy the autoregressive SJ ansatz is comparable to a conventional SJ ansatz with a single-reference mean-field wavefunction.
Is it possible to estimate the error density depend on the systems size?
As the parametrization of the Jastrow factor by a MADE network, which is not translationally invariant, scales like $N^4$ with the system size, simulation of of systems larger than $10 \times 10$ results in out-of-memory errors. The accuracy of the autoregressive model is at least as good as a simple Slater-Jastrow ansatz. Indeed, it is a pertinent question whether the direct sampling approach gives an advantage in terms of accuracy for larger system sizes compared with Markov chain MC. At present, this cannot be answered, and is left for future work.
The conclusion is rather brief. It might be useful to comment on the conclusions drawn from the benchmark and comparison to other variational methods.
In an expanded conclusion the main limitations of the current implementation are addressed. This is for one the fact that sampling and density estimation are equally costly, which makes approaches such as symmetry projection impossible. The other aspect is the imbalance between the Jastrow factor, which is heavily over-parametrized, and the Slater determinant, which is only single-reference. The outlook section has already addressed the difficulties connected with autoregressive multi-reference approaches. In conclusion, our approach is on par with simple SJ wavefunctions and is outperformed in terms of accuracy by SJ wavefunctions with symmetry projection and backflow.
Author: Stephan Humeniuk on 2023-03-14 [id 3474]
(in reply to Report 2 on 2022-12-31)We wish to thank the referee for reviewing our manuscript. We have addressed their comments and criticisms and hope that out manuscript is now suitable for publication.
The main criticism is the imbalance between the benchmark/application part and the presentation of technical details. We have extended the benchmark section with a comparison with other variational methods from the published benchmark Varbench. Indeed, at the current stage our algorithm is of proof-of-principle level and further work is needed for genuine applications. The reasons have been stated in an enlarged conclusion section: (i) The MADE network is not translationally invariant and the many parameters lead to out-of-memory errors before larger system sizes can be attained where the cubic scaling becomes relevant. (ii) The Slater determinant is of single-reference type so that the algorithm is of lower accuracy than current restricted Boltzmann machine (RBM) implementations of Slater-Jastrow wavefunctions with symmetry projection and backflow. Directions for possible improvement of our algorithm are mentioned in the outlook section.

---

## Round 1 · Referee Report · Anonymous · 2022-12-31

Strengths
- A detailed study of a VMC algorithm combining a Jastrow factor and a Slater determinant into a wave function which can be sampled directly.
- System size scaling remains cubic in system size.
- first benchmarking results are shown
Weaknesses
- the balance between technical detail level of the algorithm and the benchmarking/application part is skewed towards the former.
- the conclusion/perspective should be more clear whether this algorithm is of proof of principle level or whether it is ready for genuine applications, or whether further work is need to bring it to a level useful for applications.
Report
Given the above summary of strengths and weaknesses I believe the paper can ultimately be published in SciPost Physics after the weaknesses are addressed in a minor revision.

---

## Round 4 · Referee Report · Anonymous (Referee 3) · 2023-3-26

Strengths

- Detailed discussion of the algorithm
- Solid numerical results and benchmarks

Report

The authors addressed all concerns I raised in my previous report. I thus recommend publication in SciPost Physics.

---

## Round 4 · Referee Report · Anonymous (Referee 4) · 2023-4-2

Report

I am satisfied with the revised version submitted by the authors. The paper can now be accepted for publication in SciPost.

---

## Round 4 · Author Response

We thank both referees for their comments on our manuscript. The main criticisms from both referees
was lack of sufficient benchmark results and comparison with other variational methods. We have addressed this point by extending the benchmark section and the conclusion and hope that our manuscript is now ready for publication.
A list of changes is given below.

---

## Round 4 · List of Changes

(i) Extended benchmark section and comparison with different variants of Slater-Jastrow-RBM as published in the variational benchmark [Varbench](https://github.com/varbench/varbench) and with continuous-time QMC.
(ii) Inclusion of Hartree-Fock energies for assessing correlation energies.
(iii) The effectiveness of the cooptimization of orbitals has been assessed by starting from a non-converged (i.e. non-self-consistent) Hartree-Fock Slater determinant . The variational ground state is reached even with such a poor starting point. At variance with a statement in a previous version of the paper, cooptimization is not crucially important if the Slater determinant is taken to be a fully converged Hartree-Fock solution.
(iv) The conclusion emphasizes the limitations of the current implementation, which has proof of principle character.
(v) Explanation of dashed and dotted lines in Fig. 15.

---

## Editorial Decision

published